# Translatome analysis reveals altered serine and glycine metabolism in T-cell acute lymphoblastic leukemia cells

Kim R. Kampen [1,12], Laura Fancello [1,12], Tiziana Girardi [1], Gianmarco Rinaldi [2,3], Mélanie Planque[2,3], Sergey O. Sulima[1], Fabricio Loayza-Puch [4], Benno Verbelen[1], Stijn Vereecke [1], Jelle Verbeeck[1], Joyce Op de Beeck[1], Jonathan Royaert[1], Pieter Vermeersch[5], David Cassiman [6], Jan Cools[7,8], Reuven Agami[9,10], Mark Fiers [11], Sarah-Maria Fendt [2,3] & Kim De Keersmaecker [1]

Somatic ribosomal protein mutations have recently been described in cancer, yet their impact on cellular transcription and translation remains poorly understood. Here, we integrate mRNA sequencing, ribosome footprinting, polysomal RNA sequencing and mass spectrometry datasets from a mouse lymphoid cell model to characterize the T-cell acute lymphoblastic leukemia (T-ALL) associated ribosomal *RPL10 R98S* mutation. Surprisingly, RPL10 R98S induces changes in protein levels primarily through transcriptional rather than translation efficiency changes. Phosphoserine phosphatase (*PSPH*), encoding a key serine biosynthesis enzyme, was the only gene with elevated transcription and translation leading to protein overexpression. PSPH upregulation is a general phenomenon in T-ALL patient samples, associated with elevated serine and glycine levels in xenograft mice. Reduction of PSPH expression suppresses proliferation of T-ALL cell lines and their capacity to expand in mice. We identify ribosomal mutation driven induction of serine biosynthesis and provide evidence supporting dependence of T-ALL cells on PSPH.

[1] Laboratory for Disease Mechanisms in Cancer, Department of Oncology, KU Leuven and Leuven Cancer Institute (LKI), Herestraat 49, 3000 Leuven, Belgium. [2] Laboratory of Cellular Metabolism and Metabolic Regulation, VIB-KU Leuven Center for Cancer Biology, VIB, Herestraat 49, 3000 Leuven, Belgium. [3] Laboratory of Cellular Metabolism and Metabolic Regulation, Department of Oncology, KU Leuven and Leuven Cancer Institute (LKI), Herestraat 49, 3000 Leuven, Belgium. [4] Translational Control and Metabolism, German Cancer Research Center (DKFZ), Im Neuenheimer Feld 280, 69120 Heidelberg, Germany. [5] Department of Laboratory Medicine, University Hospitals Leuven, Herestraat 49, 3000 Leuven, Belgium. [6] Department of Gastroenterology-Hepatology and Metabolic Center, University Hospitals Leuven, Herestraat 49, 3000 Leuven, Belgium. [7] Laboratory of Molecular Biology of Leukemia, VIB-KU Leuven Center for Cancer Biology, VIB, Herestraat 49, 3000 Leuven, Belgium. [8] Laboratory of Molecular Biology of Leukemia, Center for Human Genetics, KU Leuven and Leuven Cancer Institute (LKI), Herestraat 49, 3000 Leuven, Belgium. [9] Department of Pediatric Oncology/Hematology, Erasmus Medical Center, Wytemaweg 80, 3015 CN Rotterdam, the Netherlands. [10] Division of Oncogenomics, The Netherlands Cancer Institute, Plesmanlaan 121, 1066 CX Amsterdam, The Netherlands. [11] Laboratory for the Research of Neurodegenerative Diseases, VIB-KU Leuven Center for Brain & Disease Research, Herestraat 49, 3000 Leuven, Belgium. [12]These authors contributed equally: Kim R. Kampen and Laura Fancello. Correspondence and requests for materials should be addressed to K.De k. (email: kim.dekeersmaecker@kuleuven.be)

Somatic mutations in genes encoding ribosomal proteins were recently described in 10–35% of patients with different leukemias and solid tumor types[1]. Whereas inactivating mutations and deletions in ribosomal protein L5 (RPL5, protein also known as uL18) and L22 (RPL22, eL22) are common in multiple tumor types[2–9], lesions affecting RPL10 (uL16) have mainly been described in pediatric T-cell acute lymphoblastic leukemia (T-ALL), with additional rare mutations in multiple myeloma[4,5]. RPL10 shows an intriguing mutational hotspot: almost all RPL10 mutant T-ALL patients carry the same arginine-to-serine missense mutation at residue 98 (R98S)[5,10].

We recently performed quantitative mass spectrometry on an isogenic lymphoid Ba/F3 B-cell model expressing the WT or R98S mutant allele of RPL10[11]. This allowed to compare expression levels of the 5557 most abundant proteins and revealed an upregulation of 178 (3%) and a downregulation of 68 (1%) proteins in the R98S cells ($p < 0.01$). In particular, this proteomics screen demonstrated RPL10 R98S-associated overexpression of the JAK-STAT signaling cascade and cell metabolism changes. We proposed differences in ribosomal frameshifting, proteasome activity, and JAK-STAT transcript levels as mechanisms contributing to the upregulation of the JAK-STAT cascade[11]. RPL10 R98S cells also show a cell survival advantage due to upregulation of internal ribosomal entry site (IRES)-driven translation of the anti-apoptotic factor BCL2[12]. However, a systematic genome-wide investigation of the effects of RPL10 R98S on the transcriptome and translatome has not been performed. Additionally, it is unclear to what extent previously detected quantitative proteomics changes were caused by RPL10 R98S-associated transcriptional, translational, and post-translational modulation. For this purpose, we now perform mRNA sequencing and sequencing of ribosome-associated RNA (ribosome footprinting or RPF-seq)[13] using the same isogenic Ba/F3 cell model, and integrate these datasets with our previously described quantitative proteomics and polysomal RNA sequencing datasets[11]. Ribosome footprinting provides nucleotide resolution mapping of active translation and allows to detect changes in translational efficiency (TE), based on the number of ribosome-protected fragment (RPF) reads for an mRNA, which reflects the average number of ribosomes bound to this mRNA. Polysomes (or polyribosomes) refer to multiple ribosomes bound to a single mRNA because of efficient translation. Polysomal RNA sequencing involves sequencing of polysome-attached mRNA, providing a second independent method to assess translation efficiencies. Our integrated multi-omics analyses reveal significant transcriptional changes associated with RPL10 R98S, which can explain up to 47.15% of the observed protein changes. Changes in TE are only observed for a small set of genes, including phosphoserine phosphatase (Psph). Psph, which encodes a key enzyme in serine biosynthesis, is consistently upregulated at both transcriptional and translational levels in RPL10 R98S cells, and is one of the strongest upregulated proteins associated with this mutation. RPL10 R98S cells display elevated serine and glycine biosynthesis in metabolic tracer analyses, and higher levels of these metabolites are present in conditioned culture media of RPL10 R98S cells. Interestingly, overexpression of PSPH occurs in the majority of T-ALL patient samples, and PSPH targeting can suppress human T-ALL expansion in vivo. Our results thus support dependence of T-ALL cells on the serine biosynthesis enzyme PSPH.

## Results

### RPL10 R98S induces distinct ribosome footprinting signatures.
We previously described that introduction of the RPL10 R98S mutation in lymphoid cells causes significant protein abundance changes in 4% of identified proteins[11]. These changes may be due

to gene expression regulation at the transcriptional, translational, and/or post-translational level. In order to better delineate the causes of detected protein changes in RPL10 R98S cells, we generated a ribosome footprinting dataset (sequencing of ribosome-protected mRNA fragments, RPF-seq) together with an mRNA-sequencing dataset of the same cells in this study. These two datasets were integrated with our previously published datasets of polysomal RNA sequencing and its matched mRNA sequencing, with another mRNA sequencing dataset and with the quantitative proteomics obtained from the same set of Ba/F3 RPL10 WT and R98S clones (Fig. 1a).

Ribosome footprinting was highly reproducible across three biological replicates (Supplementary Fig. 1) and ribosome footprints presented the expected length and triplet periodicity (Fig. 1b, c). The nucleotide resolution of ribosome footprinting allows investigating ribosome occupancy around the start and stop codons, but metagene plots across the most represented transcripts in the ribosome footprinting dataset did not reveal general defects in translation initiation or termination in RPL10 R98S cells (Fig. 1d). However, principal component analysis on ribosome footprints clearly separated the RPL10 R98S from WT samples (Fig. 1e).

### RPL10 R98S causes extensive transcriptional changes. Differences in ribosome footprinting signatures can be caused by altered available cellular mRNA levels (transcriptional changes), by altered numbers of translating ribosomes associated with the cellular mRNA (altered TE), or by a combination of both. We started by looking into transcriptional changes and noticed that differences in mRNA levels correlated well with differences in ribosome footprints in RPL10 R98S versus WT cells (Pearson's coefficient on log2-transformed data: 0.76) (Fig. 2a). Principal component analysis of the mRNA-sequencing dataset matching the ribosome footprinting separated the RPL10 R98S and RPL10 WT samples (Fig. 2b). The same was also observed for two additional mRNA-sequencing datasets that were previously generated from our isogenic Ba/F3 cell model (mRNA sequencing matching polysomal RNA sequencing[11] and an additional independent mRNA sequencing[12]) (Supplementary Fig. 2A). Comparing these datasets, 368 genes were consistently upregulated and 421 genes downregulated in RPL10 R98S cells, which were included for further analyses (Supplementary Data 1, Supplementary Fig. 2B). No enriched pathways were found in the upregulated mRNAs, whereas downregulated mRNAs were enriched for pathways involved in signaling, vesicle transport, cell adhesion, cell migration, and protein localization (Supplementary Data 2). We hypothesized that differences in expression levels or activity of transcription factors may play a role in the observed transcriptional changes. Therefore, we used iRegulon to predict regulators which may explain the observed differential transcriptional program in RPL10 R98S cells[14]. Many of the predicted transcription factors have a known role in hematopoiesis and leukemia development (Ikzf2, Pbx3, and Hoxa13 for upregulated genes and Fos, Nkx2-2, Gata2 for downregulated genes, Supplementary Data 3 and 4, Fig. 2c, d). Ikzf2 (Helios), a predicted regulator of upregulated transcripts, was itself overexpressed at the mRNA and protein levels, as confirmed by immunoblots in RPL10 R98S Ba/F3 cells, as well as in Jurkat T-ALL cells in which the RPL10 R98S mutation was introduced using CRISPR-Cas9 technology (Fig. 2e). Of interest, Nkx2-2 and Nkx2-1 were predicted as transcriptional regulators of the underexpressed mRNAs in RPL10 R98S cells. Both proteins have been implicated in T-ALL pathogenesis, and RPL10 R98S mutations significantly co-occur with NKX2-1 lesions in T-ALL[10,15]. However, NKX2-1 protein expression was undetectable in RPL10 WT or R98S Ba/F3

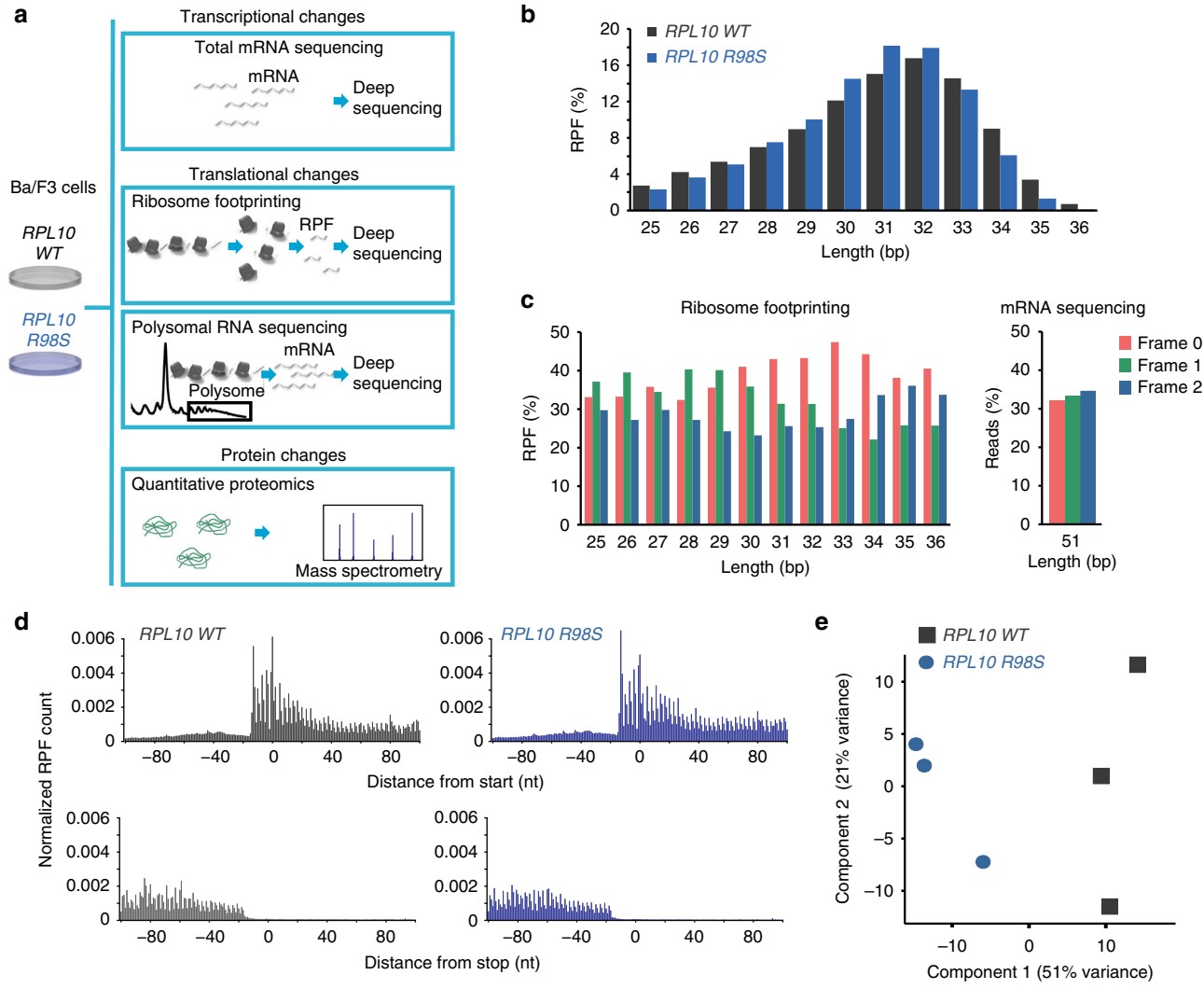

**Fig. 1** *RPL10 R98S* and *RPL10 WT* cells show distinct ribosome footprinting signatures. **a** Outline of the study design. **b** Distribution of the length of ribosome footprints (RPF, ribosome-protected mRNA fragments). **c** Left: triplet periodicity of ribosome footprinting reads; right: lack of triplet periodicity for mRNA-sequencing reads. The fraction of reads assigned to each of the three frames of translation is reported for each read length. **d** Metagene profiles of RPF densities around the start and stop codons (indicated by 0). The number of RPFs per position was averaged over all transcripts and normalized for the total number of mapped RPFs. **e** Principal component analysis based on normalized RPF counts

cells. We thus conclude that *RPL10 R98S* causes transcriptional changes of hematopoietic transcription factor target genes, with upregulation of Ikzf2 being consistently observed in two independent *RPL10 R98S* cells models.

**RPL10 R98S alters TE of an mRNA subset**. Ribosome footprinting is commonly used to estimate TE, defined as RPF counts normalized to mRNA levels. We searched for differences in TE between *RPL10 R98S* and *RPL10 WT* using the statistics provided by the Babel R package[16].

To obtain a more complete view, we also considered TE changes obtained from our previously published polysomal RNA sequencing in the same Ba/F3 cell model[11]. A total of 121 protein-coding genes showed a different TE according to ribosome footprinting and/or polysomal RNA sequencing (Supplementary Data 5 and 6, Supplementary Fig. 3A–C). Genes with insufficient sequencing coverage (<10 reads per sample in ribosome footprinting and/or polysomal RNA sequencing and in their matching mRNA) were excluded, resulting in 67 genes with a statistically significant TE change between *RPL10 R98S* and *WT* (Fig. 3a). These 67 genes showed enrichment, among

others, for cell metabolism and cell-signaling pathways (Supplementary Fig. 5D). In particular, the Jak-Stat-signaling cascade, which we previously described to be overexpressed on protein level upon *RPL10 R98S* expression[11], was enriched amongst genes with higher TE in *RPL10 R98S* cells (adjP = 2.40e−08, $n = 6$, Supplementary Fig. 3D, Supplementary Data 2). Moreover, ribosome biogenesis (KEGG, adjP = 0.0009, $n = 2$) and rRNA transcription (transcription from RNA polymerase I promoter, GO, adjP = 2.29e−02, $n = 3$) were enriched among genes with lower TE in *RPL10 R98S* cells (Supplementary Fig. 3D, Supplementary Data 2), which is in agreement with the fact that *RPL10 R98S* has previously been associated with ribosome assembly defects[5,17].

**RPL10 R98S-associated protein changes are mainly due to transcriptional modulation**. We compared the list of 67 differentially translated genes with protein measurements from our quantitative mass spectrometry dataset to verify if TE changes resulted in consistent protein level changes (Fig. 3a). For 31 out of the 67 differentially translated genes, protein measurements were available. Only Psph, belonging to the cell metabolism category of

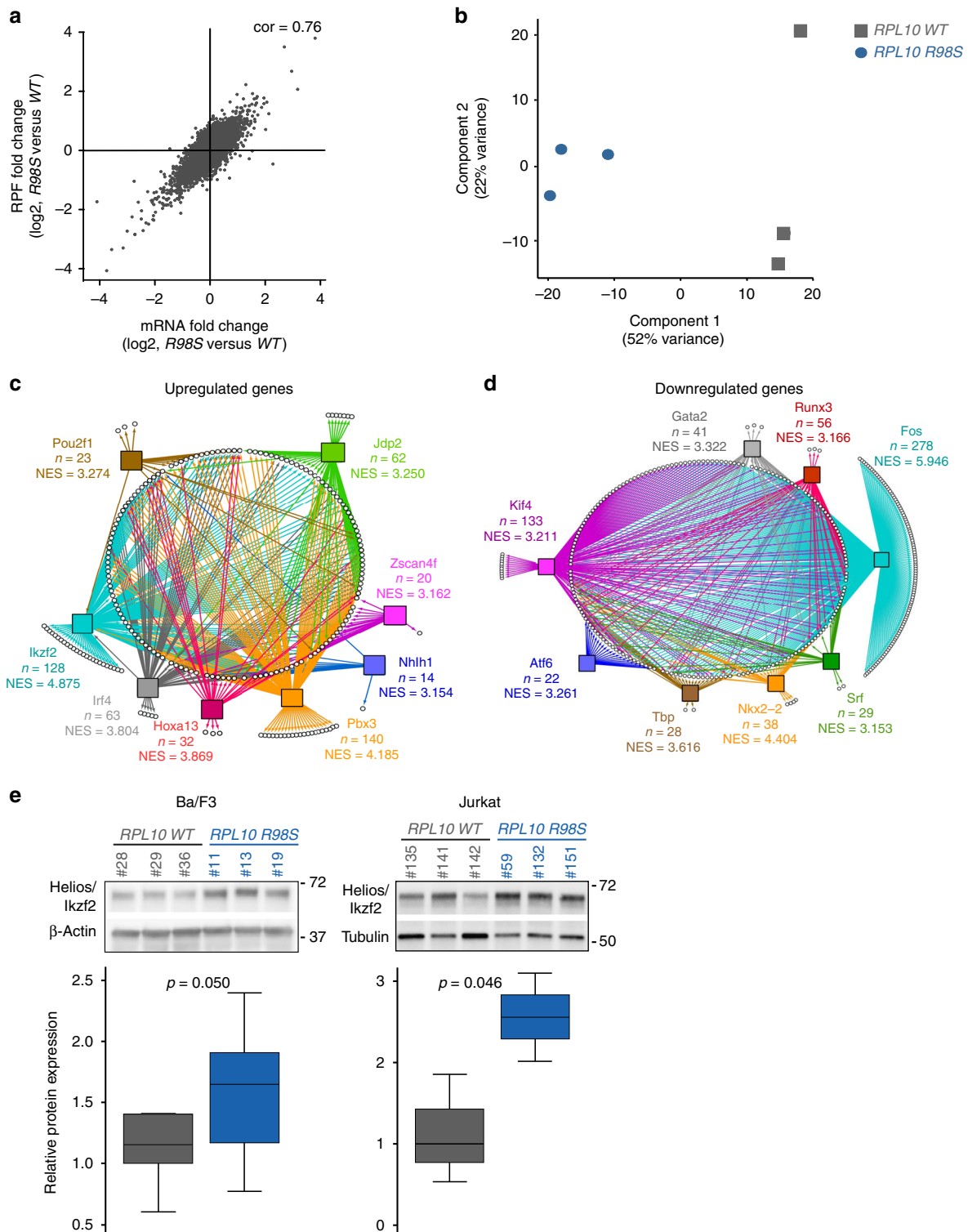

**Fig. 2** Transcriptional changes associated with *RPL10 R98S*. **a** Correlation between changes in total mRNA and RPF levels. Only genes with counts in both ribosome footprinting and matched mRNA sequencing libraries are plotted ($n = 10,645$). Reported log2-transformed fold changes were calculated by DESeq2. Cor Pearson correlation coefficient. **b** Principal component analysis based on mRNA levels (normalized read counts) from the mRNA-sequencing dataset associated with ribosome footprinting. **c, d** Network representation of transcriptionally upregulated (C) or downregulated genes (D) in *RPL10 R98S* cells. Upregulated or downregulated genes are displayed as white circles and the 8 top scoring transcription factors predicted as their regulators (iRegulon) are shown by colored squares. For each transcription factor, the number of genes that it is predicted to regulate in our mRNA-sequencing data and the normalized enrichment score (NES) are reported. A transcription factor-binding motif can be shared by several members of a transcription factor family. Only the highest scoring one as predicted by iRegulon is shown, while other transcription factors of the family may be responsible for observed mRNA expression changes. **e** Immunoblot analysis of Helios/Ikzf2 expression in *RPL10 WT* versus *R98S* expressing Ba/F3 and Jurkat cells. *P*-values were calculated using a two-tailed Student's *t*-test. All box-plots show the median and error bars define data distribution

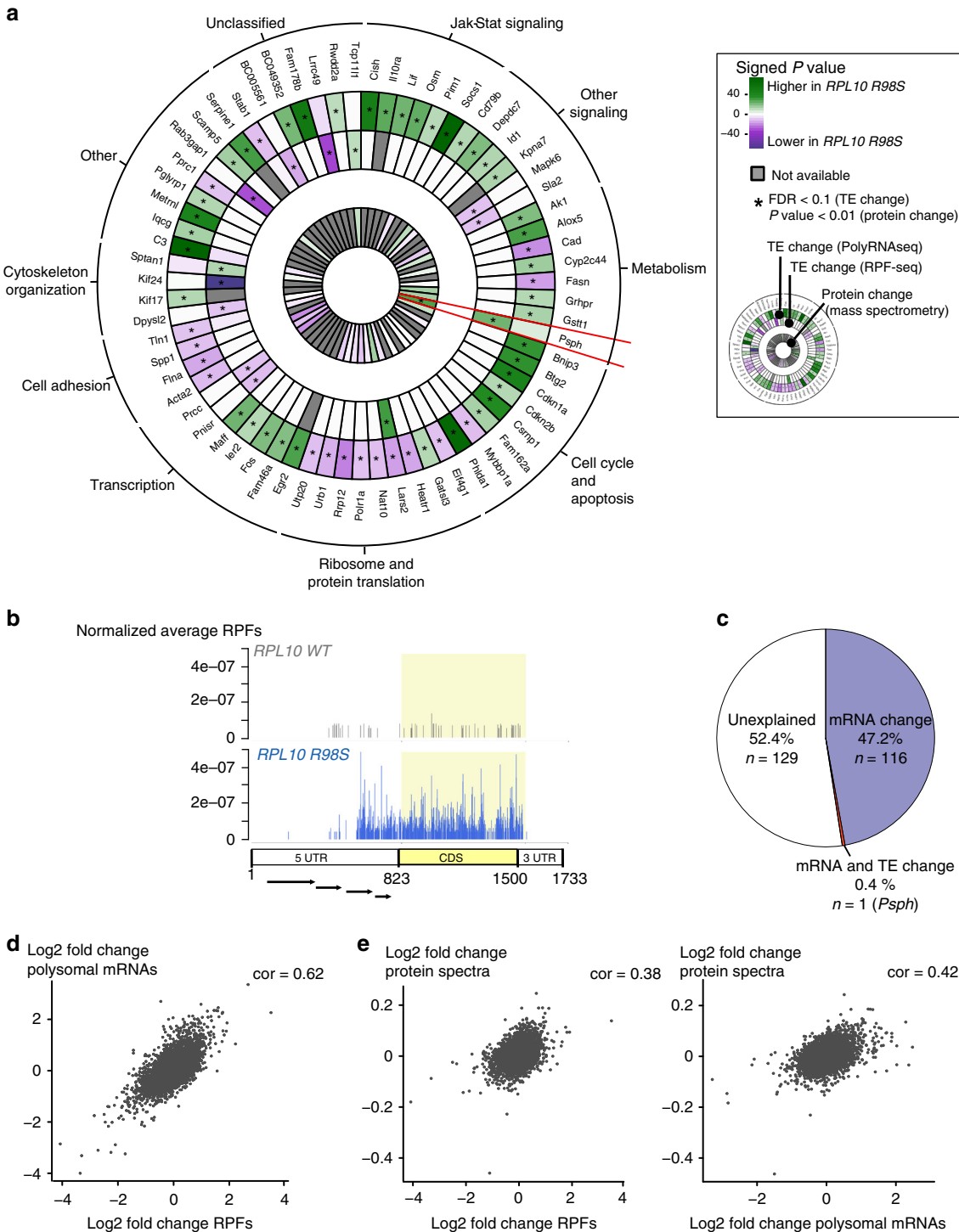

genes, showed a statistically significant protein change ($p = 0.001$, 4.7-fold upregulation) consistent with its TE change (Fig. 3a, b). As 246 proteins presented significantly altered expression levels ($p < 0.01$, $t$-test, $n = 3$ biologically independent *RPL10 WT* and *R98S* Ba/F3 clones) between *RPL10 R98S* and *WT* in the mass spectrometry dataset[11], we investigated whether the remaining differentially expressed proteins were associated with consistent changes in mRNA expression rather than TE. For *Psph* (0.4%) we observed an upregulation both in mRNA expression and in TE, whereas 47.2% ($n = 116$) of the protein changes were only associated with significant upregulation or downregulation of mRNA

expression levels, in agreement with our observation of extensive transcriptional changes associated with *RPL10 R98S* (Fig. 3c, Supplementary Data 1). For 52.4% ($n = 129$), no consistent significant change in mRNA or TE was observed (Fig. 3c, Supplementary Data 7). Linear regression models confirmed that changes in mRNA levels and in TE according to either ribosome footprinting or polysomal RNA sequencing can only explain between 36.3% and 38.9% of variability for the 246 significant protein changes ($R^2 = 0.363$ and 0.389, Supplementary Fig. 4A–E). Combining mRNA and TE changes from ribosome footprinting and polysomal RNA sequencing slightly improved

**Fig. 3** Significant TE changes identified by ribosome footprinting or polysomal RNA sequencing. **a** Circular heatmap representing the protein-coding genes with significant changes in TE (Babel, FDR < 0.1, $Z$-test with Benjamini–Hochberg correction, $n = 3$ biologically independent *RPL10 WT* and *R98S* Ba/F3 clones), identified by polysomal RNA sequencing (outer circle) or ribosome footprinting (middle circle). Corresponding protein changes, when available, are shown in the inner circle. The color scale represents the signed $p$-value associated to the change (which indicates both significance and direction of the change). Statistically significant changes are indicated by a star (*) and correspond to FDR < 0.1 for TE changes (Babel, $Z$-test with Benjamini–Hochberg correction, $n = 3$ biologically independent *RPL10 WT* and *R98S* Ba/F3 clones) and $p$-value < 0.01 for protein change ($T$-test on normalized spectra from quantitative mass spectrometry, $n = 3$ biologically independent *RPL10 WT* and *R98S* Ba/F3 clones). Only genes with at least 10 aligned ribosome footprints or polysomal RNA reads and at least 10 reads in the corresponding mRNA sequencing dataset for each sample are considered. Genes not passing this threshold or genes with no corresponding protein mass spectrometry measurement are indicated as not available. **b** Representation of the normalized RPFs for *RPL10 WT* and *R98S* Ba/F3 clones aligned to PSPH 5' untranslated region (5'UTR), coding sequence (CDS, in yellow), and 3'UTR (ENSMUST00000031399). Four arrows indicate the upstream ORFs (positions: 10–369; 373–447; 463–561; 613–681) as predicted by altORFev (10.1093/bioinformatics/btw736). These plots contain pooled data from three *RPL10 WT* versus three *R98S* Ba/F3 clones. **c** Percentages of significant protein changes (quantitative mass spectrometry, $T$-test, $p$-value < 0.01) associated with significant mRNA changes (differential expression analysis by DESeq2, two-sided Wald test with Benjamini–Hochberg correction, FDR < 0.1, $n = 3$ biologically independent *RPL10 WT* and *R98S* Ba/F3 clones) and/or with significant TE changes (Babel, $Z$-test with Benjamini–Hochberg correction, FDR < 0.1, $n = 3$ biologically independent *RPL10 WT* and *R98S* Ba/F3 clones) or neither. Both ribosome footprinting and polysomal RNA sequencing matching mRNA-sequencing datasets were considered for changes in mRNA levels. Changes in TE identified by ribosome footprinting and/or polysomal RNA sequencing were both considered. **d** Scatterplot representing the correlation between the log2-transformed fold change (*RPL10 R98S* versus *RPL10 WT*) in RPF counts and the log2-transformed fold change in polysomal RNA-sequencing counts. **e** Scatterplots representing the correlation between the log2-transformed fold changes (*RPL10 R98S* versus *RPL10 WT*) in RPF counts (on the left) or polysomal RNA-sequencing counts (on the right) and the log2-transformed fold change in normalized protein spectral counts. Cor Pearson correlation coefficient

the regression models ($R^2 = 0.403$, AIC = 502.19, Supplementary Fig. 4A+F), suggesting that these datasets provide complementary information.

The gene sets with a significant TE change, as identified by ribosome footprinting or polysomal RNA sequencing analysis, showed little overlap (Supplementary Fig. 3A). On the other hand, genome-wide changes in ribosome footprint counts and in polysomal mRNA counts correlated well (Pearson correlation on log2-transformed data: 0.62) (Fig. 3d) and each had a comparable correlation with protein changes (significant and non-significant changes considered; Pearson correlation on log2-transformed data: 0.38 and 0.42, Fig. 3e). In conclusion, combining ribosome footprinting and polysomal RNA sequencing improved our capacity to explain *RPL10 R98S* associated protein expression changes, which were primarily imposed by transcriptional rather than translational changes, along with yet undefined mechanisms.

***Psph* upregulation in *RPL10 R98S* cells induces serine/glycine synthesis**. Analysis of the *RPL10 R98S*-associated transcriptome, translatome, and proteome revealed a consistent upregulation of *Psph* transcription ($p < 0.001$, fold change (FC) = 3.8–6.5), TE (FDR = 0.006, FC = 2.5) and protein expression ($p = 0.014$, FC = 4.7) (Supplementary Fig. 5). *Psph* encodes phosphoserine phosphatase and catalyzes the last step of the serine synthesis pathway in which 3-phosphoserine is dephosphorylated to serine (Fig. 4a). *RPL10 R98S*-associated transcriptional upregulation of *PSPH* was confirmed in Ba/F3 clones and in a Jurkat T-ALL cell model expressing *RPL10 R98S* (Supplementary Fig. 6). Induction of *PSPH* TE was further supported by polysomal qRT-PCR analysis showing a shift of the distribution of ribosome-bound *PSPH* mRNA towards the most actively translated polysomal fractions in *RPL10 R98S* cells as compared to *WT* cells (Supplementary Fig. 7). These data thus collectively support increased overall PSPH translation by increased mRNA expression as well as TE. Immunoblot analysis confirmed a >10-fold and 1.5-fold upregulation of Psph protein expression in *RPL10 R98S* Ba/F3 and Jurkat clones, respectively (Fig. 4b, c, $p < 0.001$ for Ba/F3; $p = 0.032$ for Jurkat). Furthermore, we investigated Psph protein expression in hematopoietic cell cultures obtained by serial replating of lineage negative bone marrow (lin- BM) cells from *Rpl10 R98S* knock-in mice[11]. Psph was upregulated in the *RPL10 R98S* as compared to *RPL10 WT* bone marrow cells ($p = 0.004$) (Fig. 4d). To evaluate the functional impact of Psph protein

expression changes, intracellular serine and glycine concentrations were measured. The overall serine and glycine pools were significantly elevated in Ba/F3 *RPL10 R98S* clones ($p < 0.05$, Fig. 4e). Furthermore, $^{13}C_6$-Glucose tracer analysis showed that synthesis of $M + 2/M + 3$ serine and $M + 2$ glycine from $^{13}C$-Glucose was increased in *RPL10 R98S* as compared to *WT* Ba/F3 clones ($p < 0.01$ and $p < 0.05$) (Fig. 4e). Conversion of serine into glycine is a bidirectional reaction, which is catalyzed by Shmt1 in the cytosol or Shmt2 in the mitochondria (Fig. 4a). The increases in serine species containing only 2 ($M + 2$ serine) labeled carbons in the $^{13}C_6$-Glucose tracer analysis support high serine/glycine exchange in *RPL10 R98S* cells. Validation experiments showed no consistent changes in any of the enzymes involved in serine/glycine synthesis other than PSPH (Supplementary Fig. 8, Supplementary Data 8). Despite high PSPH expression in *RPL10 WT* Jurkat T-ALL cells, introduction of the *RPL10 R98S* mutation in this cell model also induced elevated total labeled serine ($M + 1$, $M + 2$, and $M + 3$ together) and glycine ($M + 1$ and $M + 2$) contribution from $^{13}C_6$-Glucose (Supplementary Fig. 9A and B, total labeled serine $p = 0.020$, total labeled glycine $p = 0.007$). Interestingly, the $^{13}C_6$-Glucose tracing also revealed significantly increased $M + 6$, and a tendency of elevated $M + 7$, $M + 8$, and $M + 9$ AMP, ADP, ATP and GMP, GDP, GTP purines, supporting incorporation of serine and glycine-derived carbons into purines (Fig. 4f, Supplementary Fig. 10). This was not observed for pyrimidine bases TMP, CMP, and UMP. These results support a general increase of PSPH protein expression in *RPL10 R98S* cell models that is associated with an enhanced de novo serine/glycine biosynthesis to generate purines. The higher serine/glycine production in *Rpl10 R98S* cells was not associated with elevated de novo protein synthesis (Fig. 4g). Serine catabolism to glycine is an already known mechanism of oxidative cancers to generate formate, which is then incorporated into purines[18,19]. In line with these findings, *Rpl10 R98S* mutant lin− BM cells presented enhanced formate levels as compared to WT lin− BM cells (Fig. 4h, $p = 0.008$). In conclusion, we show in multiple isogenic and primary cell models that *RPL10 R98S* enforces PSPH-driven serine/glycine synthesis to fuel formate and purine synthesis.

**Most primary T-ALL samples express elevated *PSPH* mRNA levels**. Next, we explored the relevance of our findings for patients with T-ALL in general, independent of the *RPL10 R98S* mutation. Analysis of mRNA-sequencing data from human T-ALL cell lines

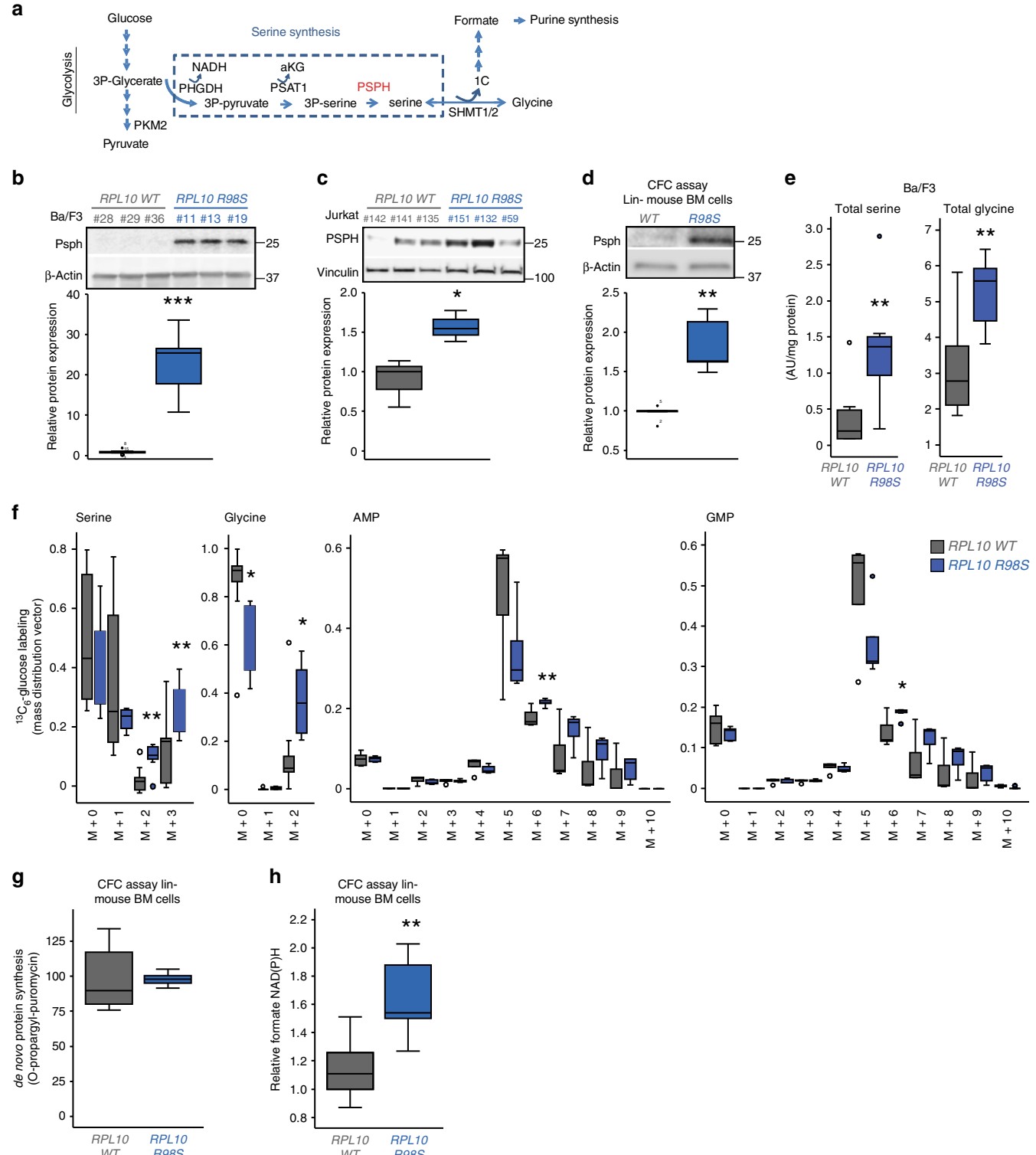

and primary patient samples[20] showed that all cell lines displayed 3- to 18-fold higher *PSPH* mRNA levels as compared to the normal thymus control (Supplementary Fig. 11A). Similarly, the majority of T-ALL patient samples showed elevated *PSPH* mRNA expression, with the two-patient samples harboring the *RPL10 R98S* mutation in the analyzed cohort showing upregulation by 8-fold (TUG7_R98S) and 13-fold (R6_R98S) (Supplementary Fig. 11A). In agreement with this, analysis of publicly available mRNA expression databases supported a general upregulation of *PSPH* in T-ALL samples, with an average 2.6-fold upregulation of

*PSPH* mRNA in T-ALL patient samples versus normal bone marrow controls (Fig. 5a). Other serine biosynthesis enzymes *PHGDH* and *PSAT1* were also upregulated by 3.2-fold and 3.4-fold in T-ALL, as opposed to glycine synthesis enzyme *SHMT2* that was unchanged between T-ALL samples and normal bone marrow controls (Fig. 5a). These analyses show that mRNA expression levels of enzymes belonging to the serine synthesis pathway are generally elevated in T-ALL, with the most significant increase for *PSPH*. In line with this finding, PSPH protein levels were elevated in primary T-ALL patient sample xenografts

**Fig. 4** Upregulation of Psph in *RPL10 R98S* cells induces serine/glycine synthesis. **a** Schematic overview of serine/glycine synthesis branching from glycolysis. **b**, **d** Immunoblot analysis of Psph in several *RPL10 WT* and *R98S* cell models: **b** Ba/F3 lymphoid cells, **c** Jurkat T-ALL cells, and **d** lineage negative (lin−) bone marrow (BM) cells. Quantifications below the blots include data from at least three biological replicates. **e** Total intracellular serine and glycine concentrations in *RPL10 WT* and *R98S* Ba/F3 cells. Eight independent Ba/F3 *RPL10 WT* clones versus six *RPL10 R98S* clones were analyzed. **f** Metabolic tracer analysis using $^{13}C_6$-glucose, measuring serine/glycine and associated tracing into purine precursors AMP and GMP. For serine/glycine measurements, we combined two independent experiments comparing eight Ba/F3 *RPL10 R98S WT* clones with six *RPL10 R98S* clones. Six independent Ba/F3 *RPL10 WT* clones versus five *RPL10 R98S* clones are analyzed for AMP and GMP. **g** Flow cytometry analysis of de novo protein synthesis by O-propargyl-puromycin (OPP) incorporation for *Rpl10 WT* and *R98S* lin− BM cells. Results from triplicate samples from two independent mouse donors are shown and relative mean fluorescent intensity (MFI) is plotted. **h** Relative formate NAD(P)H levels in the bone marrow of *WT* and *R98S* mutant mice. Formate NAD(P)H levels were subtracted from background NAD(P)H levels and corrected for protein input. The box-plots include combined results of two independent experiments comparing three *WT* versus three *R98S* mutant BM CFC assay samples, derived from independent donor mice. All box-plots show the median and error bars define data distribution. Statistical analysis *$p$-value < 0.05, **$p$-value < 0.01, ***$p$-value < 0.001. #clone IDs of Ba/F3 and Jurkat. $p$-values were calculated using a two-tailed Student's $t$-test. Color codes: gray indicates RPL10 WT control clones and blue indicates RPL10 R98S clones

as compared to normal bone marrow and acute myeloid leukemia samples (Supplementary Fig. 11B).

**Increased circulating serine and glycine levels can enhance the survival of supportive cells.** The upregulation of serine bio-synthesis enzymes in T-ALL samples encouraged us to further explore the functional contribution of serine biosynthesis in T-ALL. We measured levels of serine biosynthesis metabolites in plasma samples from mice that had reached the disease end stage upon xenografting with human pediatric T-ALL patient samples. While metabolite ion exchange chromatography revealed the elevation of (phospho-)serine in some plasma samples collected after xenografting, it was mainly glycine that was elevated (Fig. 5b). In agreement with this, conditioned media (CM) taken from *RPL10 R98S* Ba/F3 cell cultures that have reached their growth plateau contained 15–20 μM more serine and glycine as compared to CM from *RPL10 WT* cultures (Fig. 5c). $^{13}C_6$-glucose tracing analysis of conditioned media revealed that *RPL10 R98S* clones secreted more labeled serine and glycine metabolites in the medium (Fig. 5d). Moreover, the *RPL10 R98S* mutant cells showed a tendency towards less uptake of serine and glycine from the culture medium (Fig. 5e). Parental Ba/F3 cells showed a better cell survival when provided with CM from *RPL10 R98S* Ba/F3 cells (Fig. 5f). As serine was completely consumed in the media of Ba/F3 *WT* cell cultures at their growth plateau (Fig. 5c), we reasoned that concentrations of 20 μM serine might be causal for the survival benefit in *RPL10 WT* cells upon stimulation with CM from *RPL10 R98S* cells. Indeed, addition of 20 μM serine to exhausted *WT* cultures could mimic the effects observed from addition of *RPL10 R98S* CM. Moreover, the survival benefit associated with CM from *RPL10 R98S* cells could not be further enhanced by the addition of another 20 μM of serine (Fig. 5f). These data suggest that by enhancing their de novo serine/glycine synthesis, the *RPL10 R98S* cells can stimulate survival of other cells by excreting part of the synthesized serine/glycine and by reducing their uptake. This is consistent with the observation that T-ALL xenografted mice presented elevated serine/glycine levels in their blood plasma. Therefore, we reasoned that such eleva-tions in serine and glycine levels in the circulation/CM might also benefit cells that form the protective niche for the maintenance of leukemia cells. To test this hypothesis, the effects of serine and glycine addition on the survival kinetics of mouse hematopoietic cells were measured. Mouse bone marrow stromal cells showed a prolonged cell survival when serine was added, while myeloid cells showed an extended survival in the presence of glycine (Fig. 5g, $p < 0.05$ for multiple data points). Altogether, these results suggest that the upregulation of serine biosynthesis genes in T-ALL cells facilitates intrinsic de novo serine and glycine synthesis, resulting in elevated levels of these metabolites in the blood. Increased serine and glycine availability can promote the

survival of healthy cells, such as cells from the bone marrow niche, which can in turn benefit leukemia cells by providing a supportive microenvironment for engraftment and expansion.

**T-ALL cells depend on de novo serine synthesis.** Cancer cells undergo metabolic rewiring that makes them dependent on endogenously produced serine[21]. The observation that PSPH was overexpressed in our T-ALL cell model, as well as in T-ALL patient samples urged us to investigate to what extent leukemic cells are dependent on PSPH expression for their proliferation and/or survival. The effects of 40–50% reduction of PSPH protein levels by two different PSPH targeting shRNAs were explored in three independent T-ALL cell lines (Supplementary Fig. 12, Fig. 6a). PSPH knockdown reduced cell proliferation of all tested T-ALL cell lines (Fig. 6b). Consistently, non-transduced leukemic cells expanded over time, resulting in a loss of mCherry-labeled PSPH knockdown cells (Supplementary Fig. 13). DND41, the cell line with highest *PSPH* mRNA expression levels, was the least affected by PSPH knockdown, presumably due to relatively high residual PSPH levels (Fig. 6a). Apoptosis was induced in KE37, but not in RPMI8402 and DND41 (Fig. 6b). As PSPH targeting mainly affected the expansion of T-ALL cells and induced limited apoptosis, we hypothesized that PSPH knockdown interfered with one of the cell cycle checkpoints. CDK2 phosphorylation at threonine 160 (Thr-160) is required for cell cycle progression through the S-phase of the cell cycle[22], and this Thr-160 phos-phorylation was decreased in T-ALL cell lines KE37 and RPMI8402 upon PSPH knockdown (Fig. 6c). In line with these data, we observed a significant decrease in cell cycle progression in PSPH knockdown T-ALL cells (Fig. 6d, Supplementary Fig. 14A). We found evidence for increased serine catabolism to formate in *R98S* cells (Fig. 4h), and hypothesized that PSPH might be an important factor controlling catabolism to formate in order to enhance purine synthesis. Accordingly, PSPH knock-down T-ALL cells presented 20–40% reduced formate NAD(P)H levels (Fig. 6e). In contrast, de novo protein synthesis was not reduced in PSPH knockdown T-ALL cells (Fig. 6f, Supplementary Fig. 14B), suggesting that altered serine/glycine metabolism due to altered PSPH levels mainly drives formate generation and purine synthesis.

To analyze the effects of reducing PSPH levels in vivo, mice were injected with freshly transduced and >90% viable T-ALL cells containing either scrambled control or PSPH shRNA plasmids (Fig. 7A, Supplementary Fig. 15A). In vivo leukemia progression induced by KE37 T-ALL cells was comparable to that observed when injecting primary patient-derived T-ALL cells, with leukemia engraftment to the bone marrow and infiltration into the spleen (Fig. 7b+d). All KE37 xenografted animals developed leukemia. However, PSPH knockdown cells showed a significantly lower expansion potential in the bone marrow, as

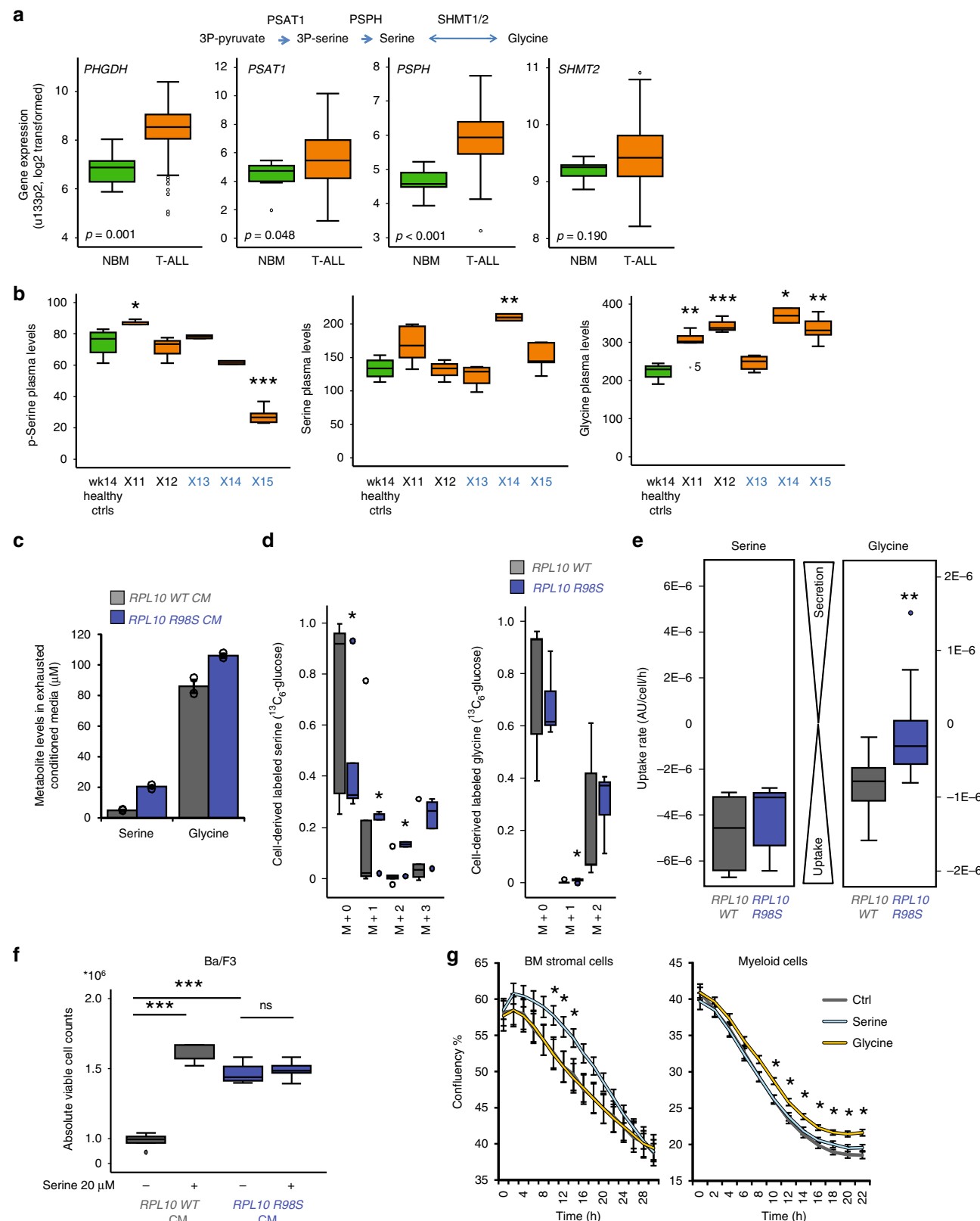

well as in the spleen in comparison to scrambled control cells, as supported by reduced percentages of mCherry-positive shPSPH containing cells for both hairpins compared to the scrambled shRNA (Fig. 7b+d). Total spleen weights of leukemic mice were significantly reduced in shPSPH#2 xenografted animals (Fig. 7c). This experiment was also performed for the RPMI8402 T-ALL cell line using shPSPH#1. RPMI8402 leukemia cells mainly infiltrated the spleen (Supplementary Fig. 15B+D), and bone marrow analysis was therefore less informative. In mice xenografted with RPMI8402, the reduction of PSPH levels also impaired the capacity of the cells to infiltrate and expand in the spleen (Supplementary Fig. 15C and D). In line with these data,

**Fig. 5** T-ALL-derived circulating serine and glycine can facilitate a cell survival benefit for leukemia supporting cells. **a** *PHGDH*, *PSAT1*, *PSPH*, and *SHMT2* Affymetrix MAS 5.0 mRNA expression levels obtained from the R2 AMC genomics analysis and visualization platform (Meijerink dataset). Data were extracted and re-plotted comparing normal bone marrow (green, NBM, *n* = 7) control samples and pediatric T-ALL samples (orange, *n* = 117). **b** Metabolite levels measured by ion exchange chromatography in plasma samples from control mice (green) and mice xenografted with the indicated T-ALL samples (orange). Metabolites are reported in μmol/L. From left to right, the boxplots represent phospho-serine, serine and glycine. X# indicates the T-ALL PDX sample ID. Control mice *n* = 4 and PDX mice X11 *n* = 5, X12 *n* = 5, X13 *n* = 5, X14 *n* = 2, X15 *n* = 5. RPL10 R98S cases are indicated in blue. **c** Ion exchange chromatography determined serine and glycine levels in pooled conditioned media from either *RPL10 WT* or *R98S* Ba/F3 clones (*n* = 2 independent biological measurements of pooled CM *n* = 3). Data are represented as mean ± standard deviation. **d** $^{13}C_6$-Glucose tracing of labeled serine and glycine released in the conditioned media of six Ba/F3 *RPL10 WT* clones versus five *R98S* clones. **e** Glycine and serine uptake rates comparing *n* = 6 Ba/F3 *RPL10 WT* clones versus *n* = 5 *R98S* clones. **f** Absolute viable cell counts of parental Ba/F3 cells to which conditioned medium (CM) taken from *RPL10 WT* or *R98S* cells was added, with or without addition of 20 μM serine. **g** Cell culture confluency plots illustrating the survival of bone marrow stromal cells and myeloid macrophages in presence and absence of 400 μM serine or glycine. Data are represented as mean ± standard deviation. All box-plots show the median and error bars define data distribution. Statistical analysis *$p$-value < 0.05, **$p$-value < 0.01, ***$p$-value < 0.001. $p$-Values were calculated using a two-tailed Student's *t*-test

X12 primary T-ALL PDX cells transduced with shPSPH#1 and shPSPH#2 were not able to expand in vivo in NSG mice, while scrambled shRNA X12 T-ALL cells were detectable in the blood, bone marrow, and spleen (Supplementary Fig. 16). These results further support that T-ALL cells depend on high PSPH expression for leukemic expansion and highlight PSPH as therapeutic target in T-ALL. Normal cells are likely not responsive to PSPH targeting due to their general low expression of PSPH and their low dependence on endogenous serine synthesis[21].

## Discussion

The impact of cancer associated somatic ribosome defects on cellular transcription and translation remains poorly understood. To address this, we generated mRNA sequencing, ribosome footprinting, polysomal RNA sequencing, and quantitative mass spectrometry datasets from an isogenic *RPL10 R98S* Ba/F3 cell model. This multi-omics approach revealed that the RPL10 R98S ribosomal defect is associated with significant transcriptional and translational changes. However, while most of the protein changes detected by mass spectrometry were supported by transcriptional regulation, ~50% of protein changes could not be explained by differences in mRNA or TE levels, suggesting post-translational regulation mechanisms. We previously identified RPL10 R98S-associated changes in proteasome composition and activity levels[11] and we speculate that differences in protein stability may explain part of these protein changes. In agreement with this, we found Jak1 among those protein changes that are unexplained by transcriptional or translational changes (Supplementary Data 7). Jak1 is a protein which we previously reported to be upregulated in *RPL10 R98S* cells due to increased protein stability[11].

An in silico analysis of *RPL10 R98S*-associated transcriptional changes identified several transcription factors involved in hematopoietic differentiation that may drive such gene expression changes. This is particularly relevant in T-ALL, which is typically associated with a block in hematopoietic T-cell lineage development due to aberrant expression of hematopoietic transcription factors[23]. Ikaros2/Helios was overexpressed in our *RPL10 R98S* Ba/F3 and Jurkat cell models, whereas inactivation of *Helios* has been described in B- and T-ALL[24,25]. However, Helios is also known to be involved in regulatory T-cell development, similar to STAT5 that was also upregulated in *RPL10 R98S* cells[26,27]. Regulatory T-cells facilitate suppression of the immune system and are therefore often increasingly found in cancers[28]. One could speculate that the observed upregulation of Helios and Stat5 expression may allow leukemic cells to suppress immune surveillance. Another predicted transcriptional regulator of

underexpressed mRNAs in *RPL10 R98S* cells was Fos. FOS can dimerize with JUN family proteins to form the AP-1 oncogenic transcription factor complex. Human RPL10 (previously named QM) and its chicken homolog (previously named Jif-1, jun interacting factor 1) negatively regulate c-JUN by inhibiting its DNA binding and transactivation[29–31]. In particular, RPL10 was reported to compete with FOS for the same binding domain on JUN[29]. These data may indicate an impact of the R98S mutation on the extra-ribosomal regulation of JUN by RPL10.

Besides transcriptional changes, our ribosome footprinting and polysomal RNA sequencing datasets also allowed to detect a subset of genes with *RPL10 R98S*-associated differences in TE. The subset of differentially translated genes identified by the two techniques showed a poor overlap, which may be due to technical and/or analytical biases. Only recently has the comparison between the two techniques been fully addressed[32]. Ribosome footprinting measures TE based on average ribosome occupancy of mRNAs whereas polysomal RNA sequencing measures this feature based on polysomal association of an mRNA. While only ribosomes associated to the canonical coding sequence are considered when estimating TE in ribosome footprinting, polycistronic mRNAs may be associated with the polysomal fraction even if ribosomes are bound to alternative rather than canonical open-reading frames (ORFs), which cannot be distinguished in polysomal RNA sequencing. Differences in sample preparation may also play a role, because polysomal RNA sequencing only uses the polysomal fraction, whereas ribosome footprinting also includes monosomes. Although polysomes are considered to contain the actively translating ribosomes, recent studies in *S. cerevisiae* revealed that monosomes are elongating and translate nonsense-mediated mRNA decay (NMD) targets, upstream ORFs, short ORFs and low abundance regulatory proteins[33]. In light of these differences, it is particularly useful to have access to both types of datasets in the same model. The fact that combining ribosome footprinting and polysomal RNA sequencing improved the prediction of protein changes, suggests that these techniques are complementary.

Serine biosynthesis enzyme PSPH showed the most consistent *RPL10 R98S*-associated change in gene expression, with transcriptional upregulation in all available mRNA datasets, increased TE according to ribosome footprinting, and increased association to the most actively translating polysomal fractions according to qRT-PCR. PSPH also represented one of the strongest upregulated proteins by mass spectrometry, and was the only gene whose changes at the transcriptional and TE level could be confirmed by a significant difference in protein expression. The serine biosynthesis pathway recently became of interest to the cancer research community, as a subset of triple negative breast cancers harbor an amplification of *PHGDH* leading to a

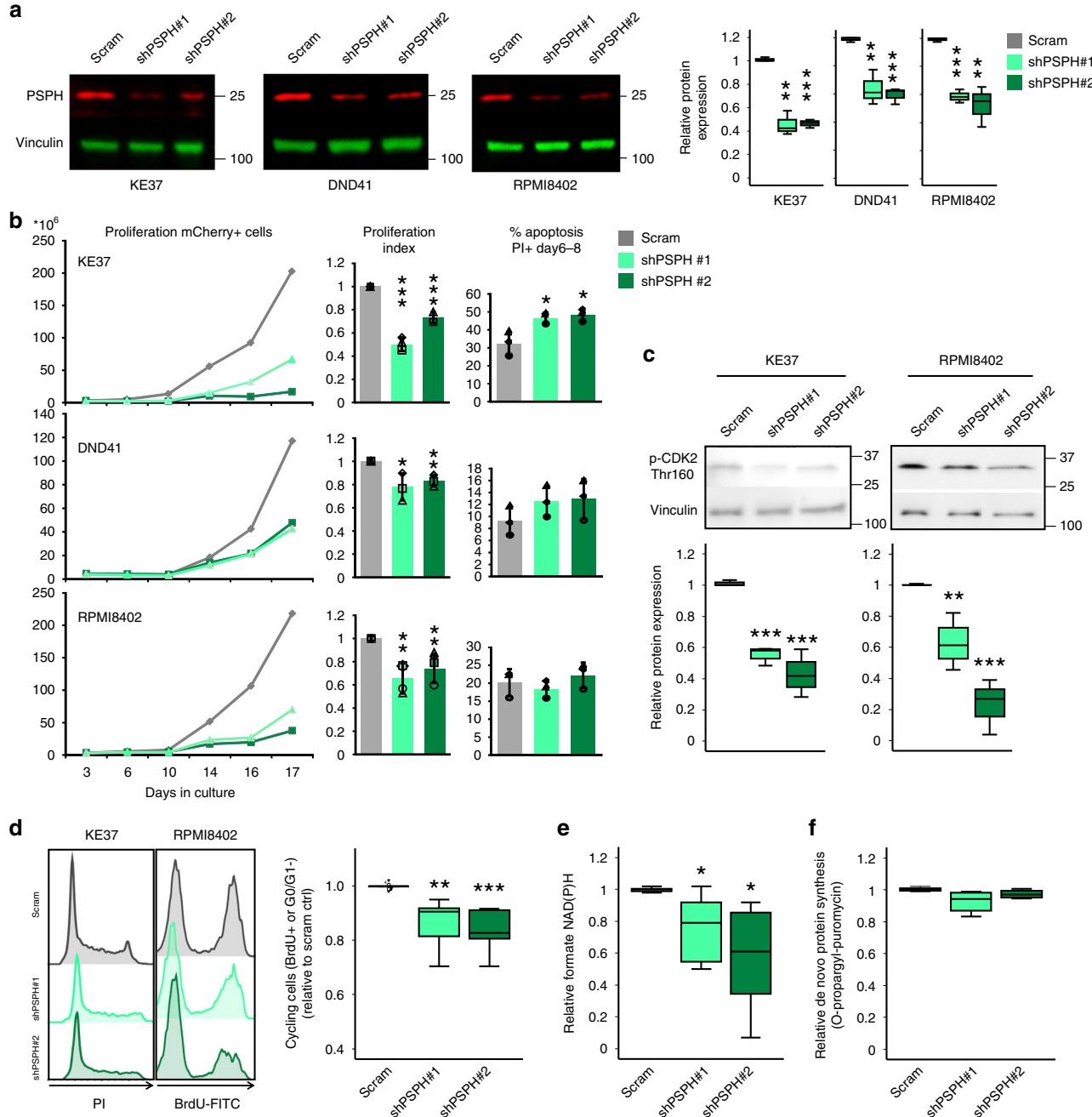

**Fig. 6** PSPH suppression blocks the expansion potential of human T-ALL cells in vitro. **a** Fluorescent immunoblot analysis of PSPH protein levels in KE37, DND41, and RPMI8402 cells upon knockdown of PSPH using two independent shRNAs, with the quantification of three independent blots on the right. **b** Left: growth curves representing proliferation of PSPH knockdown cells over time for T-ALL cell lines KE37, DND41, and RPMI8402. Middle: Proliferation index which was calculated based on pooling of at least three individual data points from the left plot in order to quantify the effects of PSPH shRNA interference on T-ALL cell proliferation. Right: Apoptosis in PSPH knockdown cells (three averaged data points). Data are represented as mean ± standard deviation. **c** Immunoblot analysis of phosphorylated CDK2 at threonine 160. **d** Left histograms: BrdU incorporation or PI cell cycle flow cytometry analysis of representative scrambled control and PSPH knockdown T-ALL cell lines. Right: Quantification of the percentage cycling cells in cultures of scrambled, shPSPH#1, and shPSPH#2 KE37, DND41, and RPMI8402 T-ALL cells. For technical reasons, some T-ALL lines were only analyzed by either BrdU or PI cell cycle analysis, and at least in two independent experiments per sample. **e** Relative formate-derived NAD(P)H levels in scrambled control and PSPH knockdown T-ALL cells (combined results for KE37, RPMI8402, DND41, X12). Background NAD(P)H levels were subtracted from formate-derived NAD(P)H levels and the data were corrected for protein input. The box-plots include combined results of two independent experiments. **f** Flow cytometry analysis of de novo protein synthesis by O-propargyl-puromycin (OPP) incorporation. Relative protein synthesis as shown in the figure was calculated as shPSPH#1 and shPSPH#2 OPP MFI relative to scrambled control cells for KE37, RPMI8402, and DND41. All box-plots show the median and error bars define data distribution. Statistical analysis *p-value < 0.05, **p-value < 0.01, ***p-value < 0.001. p-values were calculated using a two-tailed Student's t-test

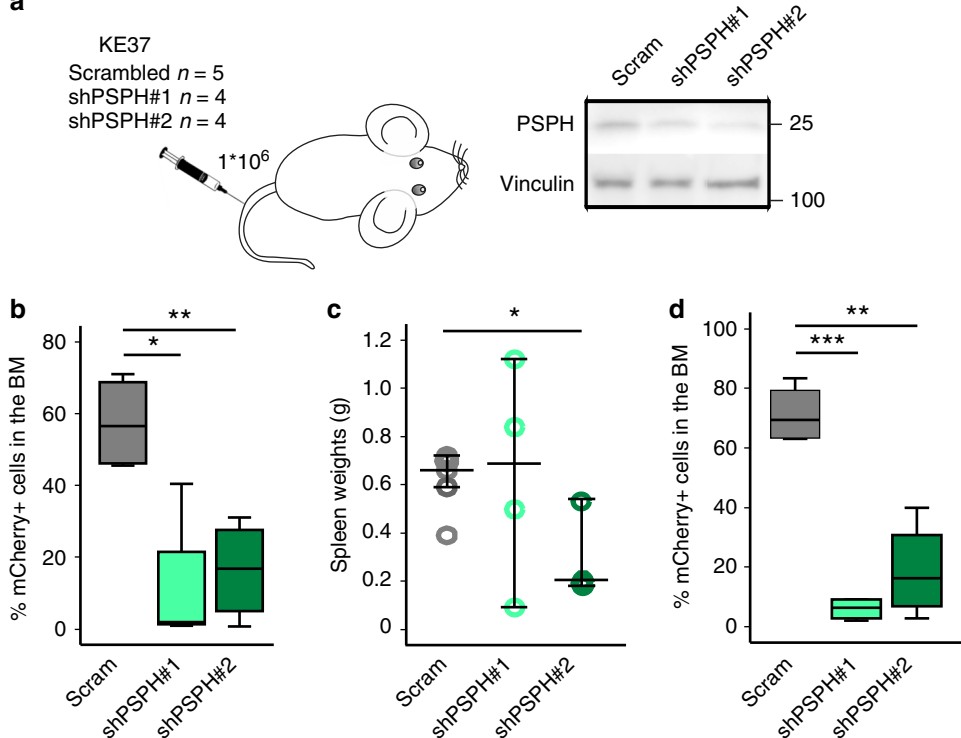

**Fig. 7** PSPH suppression blocks the expansion potential of human T-ALL cells in vivo. **a** Left: Schematic experimental overview of the in vivo set-up to test the effects of PSPH knockdown on leukemia engraftment and progression by tail vein injection of 1*10[6] scrambled control, shPSPH#1, or shPSPH#2-transduced cells 24 h after initial transduction. Right: immunoblot confirmation of PSPH knockdown in the cells that were injected in the mice. **b** Percentage of mCherry-expressing cells in the bone marrow of leukemic mice as determined by flow cytometry. **c** Spleen weights of the leukemic mice at disease end stage. Data are shown as individual data points accompanied by the median and whiskers representing data distribution. **d** Percentage of mCherry-expressing cells in the spleen of leukemic mice as determined by flow cytometry. All box-plots show the median and error bars define data distribution. Statistical analysis: *p-value < 0.05, **p-value < 0.01, ***p-value < 0.001. p-values were calculated using a two-tailed Student's t-test

corresponding overexpression of this serine biosynthesis enzyme[34,35]. *PHGDH* amplified breast cancer cell lines are dependent on de novo serine synthesis for their proliferation[36]. High tumor cell-specific PSPH expression was observed in hepatocellular carcinoma and associated with inferior patient survival[21]. Constitutive PSPH expression has been shown to induce tumor progression in an in vivo model of hepatocellular carcinoma, while shRNA targeting of PSPH reduced the tumor burden in the same model[21]. Here, we show that de novo serine/glycine synthesis is increased via PSPH upregulation upon the introduction of the T-ALL-specific *RPL10 R98S* mutation. Conditioned medium of *RPL10 R98S* cells contained higher serine and glycine levels, and we showed that exposure of wild type cells to extra serine or glycine promotes their proliferation and survival. Whilst the lack of a dose-dependent effect of serine addition to RPL10 R98S-conditioned Ba/F3 medium does not necessarily imply that another factor is involved, this remains a possibility. Furthermore, our results support that enhanced reversible serine/glycine turnover within the leukemia cells mainly functions to fuel formate generation, which can serve for purine synthesis of T-ALL cells (Fig. 8). In contrast to naïve T-cells, activated T-cells also show increased serine catabolism into formate for purine synthesis[37]. While normal T-cells are mainly dependent on exogenous uptake of serine/glycine from the environment, *RPL10 R98S* cells show a high dependence on PSPH-driven endogenous serine/glycine synthesis for the generation of formate to fuel purine synthesis[37]. High expression levels of PSPH was not unique to RPL10 R98S-positive T-ALL samples, supporting that other unknown mechanisms contribute to high PSPH expression in T-ALL. Therefore, the RPL10 R98S mutation may not be the

sole event identifying patients that may benefit from PSPH inhibition.

In addition to PSPH upregulation, we describe a general induction of serine synthesis enzymes in T-ALL samples. PHGDH and PSPH are not located in regions that are recurrently amplified in T-ALL[10,38]. However, increased serine synthesis has been described in Cyclin D3:CDK4/6 complex driven T-ALL as a side effect from inhibition of glycolysis enzymes 6-phosphofructokinase (PFKP) and pyruvate kinase M2 (PKM2), causing redirection of glycolytic intermediates into the pentose phosphate (PPP) and serine pathways[39]. Targeting the serine pathway in T-ALL may represent an attractive therapeutic approach. For example, it has previously been shown that the CDK4/6 inhibitor palbociclib indirectly inhibits the serine pathway by increasing the endogenous activity of PFKP and PKM2[39]. In our *RPL10 R98S* model, it can be hypothesized that cells might be sensitive to palbociclib-induced activation of PFKP and PKM2 to counteract the enhanced expression and activity of the PSPH-driven serine synthesis in this T-ALL subset.

In conclusion, we describe the importance of PSPH in human T-ALL, which can be transcriptionally and translationally upregulated by the introduction of the ribosomal *RPL10 R98S* mutation. Our data emphasize a regulatory function of a ribosomal protein mutation in the metabolic rewiring of leukemic cells.

## Methods
**Datasets**. All RNA sequencing and proteomics datasets were generated from isogenic mouse lymphoid Ba/F3 cells engineered to express the *WT* or *R98S* mutant allele of the human *RPL10* gene[11], with three independent *WT* and three *R98S* cell clones analyzed in each experiment.

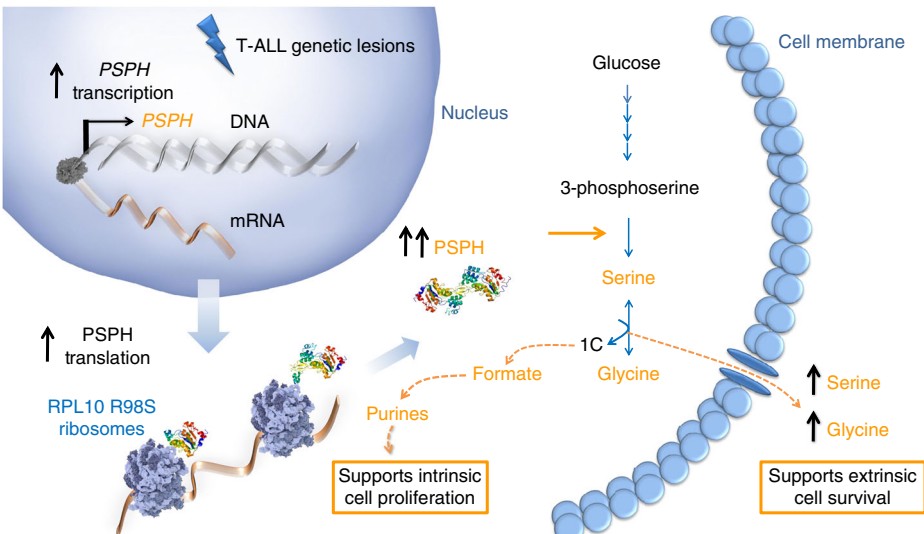

**Fig. 8** Schematic overview of PSPH upregulation in T-ALL. T-ALL cells display increased PSPH expression, which is mediated by transcriptional and translational upregulation in cells with the *RPL10 R98S* mutation. Overexpression of PSPH promotes serine/glycine production that supports the survival of neighboring cells. Moreover, serine catabolism fuels formate production and subsequent purine synthesis that enhances proliferation of the leukemic cell itself

**Polysomal RNA sequencing and matched total mRNA sequencing.** Up to three polysome associated and matched total mRNA sequencing libraries were generated for each of the three monoclonal Ba/F3 cultures expressing either *WT* or *R98S RPL10*[11]. Briefly, polysomal fractions (fractions containing at least two ribosomes, see Supplementary Fig. 17) were pooled and RNA was extracted using the phenol/chloroform method with inclusion of an extra washing step with 70% ethanol. Libraries were generated from total mRNA and polysome bound RNA using the TruSeq Stranded mRNA Sample Prep Kit (Illumina) and were sequenced on an Illumina 500 instrument.

**Supplementary total mRNA sequencing dataset.** Five million cells were treated with cycloheximide (CHX, 100 µg/ml) for 5 min, followed by RNA isolation using the Maxwell 16 LEV simplyRNA Cells Kit (Promega). Sequencing libraries were generated using the TruSeq Stranded mRNA Sample Prep Kit (Illumina) and were sequenced on an Illumina HiSeq 2500 instrument.

**Ribosome footprinting and matched total mRNA sequencing.** Two hundred million cells were treated with CHX (100 µg/ml) for 5 min and lysed in buffer (20 mM Tris–HCl, pH 7.8, 100 mM KCl, 10 mM MgCl₂, 1% Triton X-100, 2 mM DTT, 100 µg/ml CHX, and complete protease inhibitor). Lysates were centrifuged at 5200× *g* and the supernatant was digested with 2 U/µl of RNase I (Thermo Scientific) for 40 min at room temperature. Lysates were fractionated on a linear sucrose gradient (7–47%) using the SW-41Ti rotor at 28,3807×*g* for 2 h. Fractions enriched in ribosomes were pooled and treated with proteinase K (Roche) in a 1% SDS solution. Released RNA fragments were purified using Trizol. For library preparation, RNA was gel-purified on a denaturing 10% polyacrylamide urea (7 M) gel. A section corresponding to 30–33 nucleotides was excised, eluted, and ethanol precipitated. The resulting fragments were 3′-dephosphorylated using T4 PNK (New England Biolabs) for 4 h at 37 °C in 2-(N-morpholino)ethanesulfonic acid (MES) buffer (100 mM MES–NaOH, pH 5.5, 10 mM MgCl2, 10 mM β-mercaptoethanol, 300 mM NaCl). 3′ adaptor was added with T4 RNA ligase 1 (New England Biolabs) for 2.5 h at 37 °C. Ligation products were 5′-phosphorylated with T4 polynucleotide kinase for 30 min at 37 °C. 5′ adaptor was added with T4 RNA ligase 1 for 18 h at 22 °C. The resulting fragments were incubated with a biotinylated ribosomal RNA (rRNA) oligo pool (10 µM each, see Supplementary Data 9) for 10 min at 37 °C, upon short denaturation at 95–100 °C for 1 min, followed by rRNA depletion using MyOne Streptavidin C1 DynaBeads (Thermo Fisher Scientific). The resulting rRNA-depleted fragments were reverse transcribed using the SuperScript III cDNA synthesis kit. Samples were PCR-amplified for 15 cycles and the products were multiplexed and sequenced on an Illumina HiSeq 2500 platform. Matched total mRNA samples were processed as described in the previous section.

**Pre-processing and genome mapping.** Reads from Polysomal RNA sequencing and matched mRNA sequencing libraries[11] were processed as following. The first 5′ nucleotide was removed using seqtk (https://github.com/lh3/seqtk), as it frequently represents an untemplated addition from reverse transcription[40]. For ribosome footprinting libraries, adapters were trimmed using fastq-mcf (https://code.google.com/p/ea-utils/), and only clipped reads of a minimum length of 20

nucleotides were kept. rRNA and tRNA contamination was computationally removed by mapping to tRNA and rRNA references using Bowtie2[41] and collecting only unaligned reads. These reads were then aligned to the mouse mm10 reference genome and to spliced transcripts using Tophat v2.0.11[42]. Only primary alignments with mapping quality (mapqual) ≥ 10 were retained. For ribosome footprinting, only reads aligned to the CDS and excluding those aligned to the first 15 and last five codons due to the accumulation of ribosomes[43,44] were considered for feature counting. For reads from mRNA sequencing, feature counting was performed considering reads aligned to exons.

**Analysis of differentially expressed transcripts.** DESeq2[45] was applied on the mRNA datasets to identify differentially expressed transcripts between *R98S* and *WT* (FDR < 0.1). The genes with consistent and significant differential expression in all three available mRNA datasets (supplementary mRNA, mRNA matched to the polysome profiling, and mRNA matched to ribosome footprinting) were considered for enrichment analyses and used as input for iRegulon[14] to identify transcription factors that may act as key regulators of the identified downregulated and up-regulated mRNAs.

**Metagene analyses around start and stop codons.** Metagene profiles of ribosome footprint densities were generated for each sample around the start and stop codons, including 100 bp upstream and downstream. Only transcripts with at least 256 aligned ribosome footprints and having a known 5′ UTR and 3′ UTR were considered. Each read was represented by the most 5′ mapped nucleotide and read counts per position were averaged over all transcripts and normalized for the number of mapped reads.

**Ribosome footprint density profiles.** Ribosome footprint density profiles along the mRNAs of interest were generated using alignments from transcriptome mapping. The position of the most 5′ mapped nucleotide for each read is shown. The number of ribosome footprints per position was normalized for the transcriptome mapped library size and averaged between three replicates per condition. Transcriptome mapping was performed using a custom non-redundant transcriptome containing only the longest isoform of validated or manually annotated protein-coding transcripts for each gene. The Bowtie[46] mapping program was used in three steps with progressively lower stringency in order to optimize the number of mapped reads. First, only uniquely aligned reads with no mismatches in the seed were retained; then, remaining reads were mapped again and uniquely mapping reads with one mismatch in the seed were retained; finally, the remaining unmapped reads were mapped allowing for one mismatch in the seed and non-unique alignment.

**TE calculation.** The abundance of ribosome footprints of an mRNA depends on the translation rate and on the level of mRNA expression. Therefore, TE is commonly estimated as the ratio between ribosome footprint counts (ribosome protected fragments, RPF) and expressed mRNA counts (mRNA) (TE = RPF/mRNA). Differences in TE between *R98S* and *WT* conditions were calculated as TE(*R98S*)/TE(*WT*). The Babel R package[16] was used to estimate the statistical significance of

TE differences between *WT* and *R98S*. TE calculations on polysomal RNA sequencing were described previously[11].

**Enrichment analyses**. Enrichment for pathways (KEGG) or Gene Ontology functional categories (GO Biological Process Non-Redundant) was calculated using WebGestalt[47] with a minimum of two genes and with *p*-adj < 0.1 (hypergeometric test with Benjamini–Hochberg correction) for genes showing significant changes in transcription or TE between the RPL10 R98S and WT condition.

**Cell cultures**. RPL10 R98S positive leukemia cells only express mutant RPL10[5]. To mimic this, Ba/F3 cells (DSMZ) were transduced with retroviral vectors encoding human WT or R98S RPL10 cDNAs and endogenously expressed Rpl10 was knocked down with an Rpl10-targeting shRNA. Liquid cultures were established from single cell colonies grown in Clonacell-TCS medium (Stemcell technologies) followed by selection of cultures with ≥ 90% knock-down of endogenous Rpl10 as determined by qPCR. Expression of RPL10 R98S and knock-down of endogenous Rpl10 were confirmed by Sanger sequencing of cDNA. Cells derived from three monoclonal RPL10 WT or R98S-expressing Ba/F3 cultures were used for all the experiments. For Ba/F3 media exchange experiments, cells were cultured for 5 days without medium exchange or sub-culturing of the cells (overgrowth condition). At that point, half of the culture media was replaced by either *RPL10 WT*-conditioned media or *RPL10 R98S*-conditioned media in the presence or absence of 20 μM serine for 2 days, after which the cultures were analyzed for viable cell counts on a MACSQuant VYB (Miltenyi) by propidium iodide (PI) exclusion.

To set up hematopoietic cell cultures from lineage negative bone marrow cells, Rpl10 R98S conditional knock-in mice[11] were crossed to Mx1-Cre C57Bl/6 mice (B6.Cg-Tg(Mx1-cre)1Cgn/J strain Jackson Laboratories). Lineage-negative cells were isolated (EasySep Mouse Hematopoietic Progenitor Cell enrichment kit, Stemcell Technologies) from 6 to 8 weeks old Mx1-Cre Rpl10cKI R98S and from Rpl10cKI R98S control mice. Cells were plated at 2000 cells/ml in Methocult GF M3534 medium (Stemcell Technologies) containing 1250 units/ml of IFNβ (R&D systems) to induce recombination of the conditional Rpl10 R98S allele. After 10–15 days, cells were replated in fresh Methocult GF M3534 medium or analyzed as indicated. Expression of the R98S mutation upon Cre recombination was confirmed by Sanger sequencing of cDNA of the region encoding Rpl10 R98.

Jurkat, DND41, RPMI8402, and KE37 T-ALL cell lines were obtained from Leibniz-Institute DSMZ and grown in RPMI-1640 with 20% fetal calf serum. Used cells were mycoplasma negative. RPMI8402 and DND41 were authenticated by confirming unique NOTCH1 mutational status.

Jurkat cells carrying the *RPL10 R98S* mutation were generated by CRISPR/Cas9 genome editing. Jurkat cells were transduced with lentiCRISPR-Cas9, a lentiviral Cas9 encoding plasmid. These Jurkat cells were then electroporated (6 square wave pulses, 0.1 ms interval, 175 V) with a pX321 vector[48] containing an RPL10 targeting gRNA (5′-TCTTGTTGATGCGGATGACG-3′) and with a 127-nt donor oligo encoding the RPL10 R98S allele, as well as three silent mutations to avoid re-recognition and cutting by the gRNA-Cas9 complex (5′-CCTGTCAGCCCCAGC ACAGGACAACATCTTGTTAATGCTGATCACGTGAAAGGGGTGGAGCCGC ACCCGGATATGGAAGCCATCTTTGCCACAACTTTTTACCATGTACTTATT GGCACAAATTCGGGCA-3′; Integrated DNA Technologies). Following electroporation, cells were incubated for 48 h in the presence of 500 nM SCR7 (Sigma-Aldrich), followed by single cell sorting (BD FACSAria I) into 96-well plates. Growing clones were expanded and screened for the desired modification using a PCR approach with primers distinguishing the *WT* from the *R98S* allele (Fw_WT: 5′-CTTCCACGTCATCCGCATC-3′; Fw_R98S: 5′-CCTTTCACGTGAT CAGCATT-3′; Rev_WTandR98S: 5′-GCTCTGATAAAATAATGCAA GCCTA-3′). Finally, *RPL10* mutational status was confirmed by Sanger sequencing.

**Lentiviral vectors**. shRNA sequences targeting PSPH (#1 ATTCTACTTTAACG GTGAATA; #2 AGTCGCCTACAGGAGCGAAAT) or scrambled control were inserted into a pLKO1-mCherry vector (Supplementary Fig. 12A). Lentiviral particles were generated by co-transfection of the pLKO1-mCherry plasmids with packaging plasmid psPAX2 and the envelope plasmid pMD2.G (VSV-G) into Hek293T cells using GeneJuice Transfection Reagent (Merck Millipore). Leukemic cells were incubated overnight with lentiviral supernatants after which stably transduced cells were expanded and injected into NSG mice. Transduction efficiencies reached above 80% (Supplementary Fig. 12B).

**13C Tracer analysis**. Labeling experiments were performed in glucose-free RPMI supplemented with $^{13}C_6$-glucose to a final concentration of 11.1 mM and with 10% dialyzed serum for 48 h. $^{13}C_6$-glucose tracer was purchased from Sigma-Aldrich. Fifty milliliters of falcon tubes with methanol 60% and 10 mM ammonium acetate were precooled in a dry ice–ethanol bath (−40 °C) for 20 min. Cells in suspension were transferred with a 1000 μl pipette from the culture plates to 15 ml falcon tubes and pelleted by centrifugation (300×*g*, 3 min, room temperature) to remove excess of medium. Medium was aspirated and the remaining cell pellet was immediately added to the 50 ml falcon tube containing 5 ml of the quenching solution at −40 °C. Cells were pelleted by centrifugation (600×*g*, 30 s, room temperature) and the supernatant was removed by quickly inverting the tube. The cell pellet was

washed a second time with quenching solution (−40 °C). After the second wash, the cell pellet was stored at −80 °C.

Metabolites extraction was achieved by a cold two-phase methanol–water–chloroform extraction. The samples were resuspended in 800 μl of precooled 60% methanol followed by addition of 500 μl of precooled chloroform. Samples were vortexed for 10 min at 4 °C and then centrifuged (max. speed, 10 min, 4 °C). The methanol–water phase containing polar metabolites was separated and dried using a vacuum concentrator at 4 °C overnight. The samples were stored at −80 °C. Polar metabolites were analyzed by GC–MS and LC–MS.

For GC–MS measurement, polar metabolites were derivatized with 20 mg ml$^{-1}$ methoxyamine in pyridine for 90 min at 37 °C and subsequently with N-(tert-butyldimethylsilyl)-N-methyl-trifluoroacetamide, with 1% tert-butyldimethyl chlorosilane for 60 min at 60 °C. Metabolites were measured with a 7890A GC system (Agilent Technologies) combined with a 5975C Inert MS system (Agilent Technologies). One microliter of sample was injected onto a DB35MS column with a split ratio1 to 3, and an inlet temperature of 270 °C. The carrier gas was helium with a flow rate of 1 ml min$^{-1}$. For the measurement of polar metabolites, the GC oven was held at 100 °C for 1 min and then ramped to 105 °C at 2.5 °C/min and with a gradient of 2.5 °C/min finally to 320 °C at 22 °C/min. The MS system was operated under electron impact ionization at 70 eV and a selected-ion monitoring (SIM) mode was used for the measurement of metabolites. Mass distribution vectors were extracted from the raw ion chromatograms using a custom Matlab M-file, which applies consistent integration bounds and baseline correction to each ion. Data were corrected for naturally occurring isotopes[49].

For the detection of metabolites by LC–MS, a Dionex UltiMate 3000 LC System (Thermo Scientific) with a thermal autosampler set at 4 °C, coupled to a Q Exactive Orbitrap mass spectrometer (Thermo Scientific) was used for the separation of polar metabolites. After the injection of 15 μl of sample, the separation of metabolites was achieved with a flow rate of 0.25 ml/min, at 40 °C, on a C18 column (Acquity UPLC HSS T3 1.8 μm 2.1 × 100 mm). A gradient was applied for 40 min (solvent A: 0 $H_2O$, 10 mM tributyl-amine, 15 mM acetic acid—solvent B: Methanol) to separate the targeted metabolites (0 min: 0%B, 2 min: 0%B, 7 min: 37%B, 14 min: 41%B, 26 min: 100%B, 30 min: 100%B, 31 min: 0%B; 40 min: 0%B).

The MS operated in negative full scan mode (*m/z* range: 70–1050 and 300–800 from 8 to 25 min) using a spray voltage of 4.9 kV, capillary temperature of 320 °C, sheath gas at 50.0, auxiliary gas at 10.0. Data was collected using the Xcalibur software (Thermo Scientific) and analyzed with Matlab for the correction of natural abundance and to determine the isotopomer distribution, using the same method as described above for the analysis of GC–MS data.

**Formate assays**. Cellular formate levels were determined according to manufacturer's protocol (Formate assay, Biovision). In this assay, formate is oxidized and resulting NAD(P)H levels generate a colored product that was measured on a VICTOR platereader (Perkin Elmer) at λ = 450 nm to quantify formate. Correction for protein input was done using the Bradford method and readings of background NAD(P)H levels were subtracted from all samples.

**Flow cytometry**. Proliferation was measured by counting viable cells over time (growth curve), and cell cycle was analyzed by bromodeoxyuridine (BrdU) incorporation and/or propodium iodide (PI) staining. BrdU incorporation was measured following 70% EtOH fixation, HCl denaturation, EDTA neutralization, and BrdU-FITC antibody staining (Thermo Fisher). PI cell cycle analysis required 70% EtOH fixation followed by 1 h incubation with PI solution (25 μg/mL PI, 300 μg/ml RNAse, 0.05% Triton-X100, Sigma Aldrich) at 37 °C. Protein synthesis was assessed using O-propargyl-puromycin (OPP) incorporation according to manufacturer's protocol with the adaptation of methanol fixation (Click-iT, Thermo Fisher). All samples were measured using a MACSQuant VYB (Miltenyi) flow cytometer and FlowJo software.

**Metabolite measurements**. Plasma samples collected from T-ALL grafted NSG mice at disease end stage were analyzed by ion chromatography to determine metabolite amounts in the blood plasma. Conditioned media from cell cultures were filtered (0.45 μm) and analyzed by ion chromatography to measure metabolite leftovers in the media of cells in overgrowth condition. Ion chromatography was performed according to diagnostics criteria at Laboratory Medicine, University Hospitals Leuven, UZ Leuven, Belgium.

**Effects of serine/glycine on stromal cells and macrophages**. Primary C57BL/6 mouse cells were extracted from the bone marrow (bone marrow stromal cells) by one day adherence to the culture flasks and myeloid macrophages were extracted by peritoneal lavage. Recovered cells were plated in 96-well dishes in six technical repeats per condition. Cells were cultured in RPMI-1640 containing 5 μM serine and 85 μM glycine with or without the addition of 400 μM serine or glycine, and effects on cell survival were monitored by imaging adhering cells on an IncuCyte Zoom system (Essen Bioscience), as apoptotic cells shrink and detach.

**Immunoblotting**. Cells were lysed (cell lysis buffer, Cell Signaling Technology) and denatured in 1x Laemmli sample buffer (Bio-rad) containing 2-mercaptoethanol (Sigma Aldrich). Proteins were separated on Criterion Tris-Glycine eXtended gels

(Bio-rad), transferred to PVDF membranes using a Trans-Blot Turbo system (Bio-rad), incubated with primary antibodies targeting PSPH (Bioconnect, 14513-1-AP, 1:1000), phospho-CDK2 (4539, 1:1000) or Helios (42427, 1:1000) (Cell Signaling Technology), and with secondary Goat Anti-Rabbit (31462, 1:2000) or Goat Anti-Mouse (31432, 1:2000) IgG-HRP antibody (ThermoFischer Scientific) or near infra-red anti-Mouse (5470, 1:1000) or anti-Rabbit (5151, 1:1000) antibodies (Cell Signaling Technology). Proteins were visualized using fluorescent secondary antibodies or chemiluminescence on an Azure C600 (Azure Biosystems). Quantification was performed using LI-COR Image Studio Lite software version 5.2. Vinculin (V9131, 1:4000) or β-actin (A 1978, 1:4000) (Sigma Aldrich) were used to normalize for protein input. Immunoblots were repeated at least three times and representative blots are shown in the figures and in full size in Supplementary Fig. 18.

**Polysomal qRT-PCR analysis**. Cell lysates from three Ba/F3 *RPL10 WT* and three *R98S* clones were applied on a sucrose gradient and centrifuged in order to get a distribution of monosomes and polysomes. An amount of $15 \times 10^6$ cells were pelleted by centrifugation (5 min, 470×g) and were lysed in ice-cold 100 mM KCl, 20 mM Hepes (Life technologies), 10 mM MgCl₂, 1 mM DTT, 1% sodium deoxycholate, 1% NP-40 (Tergitol solution, Sigma-Aldrich), 100 μg ml$^{-1}$ cycloheximide, 1% Phosphatase Inhibitor Cocktail 2 (Sigma-Aldrich), 1% Phosphatase Inhibitor Cocktail 3 (Sigma-Aldrich), 1% Protease Inhibitor Cocktail (Sigma-Aldrich), 100 U ml$^{-1}$ RNasin (Promega). After 10 min incubation on ice, lysates were centrifuged 5 min at 15,871×g and the resulting supernatant was loaded onto 10–60% sucrose density gradients (100 mM KCl, 20 mM Hepes, 10 mM MgCl₂). Gradients were then centrifuged in a SW40Ti rotor (Beckman Coulter) at 28,3807×g for 150 min and polysomal fractions were monitored through a live OD254 nm measurement on a BioLogic LP System (Bio-Rad). Eleven fractions of equal volume were collected. For technical reasons, the three fractions corresponding to the highest polysomes were subsequently combined and considered as one fraction. RNA was extracted from equal volumes of the five remaining fractions using the phenol–chlorophorm method including an extra washing step with 70% ethanol followed by cDNA synthesis. On each fraction, qRT-PCR was performed for *Psph* in quadruplicate.

**Quantitative RT-PCR**. RNA was extracted using the RNeasy kit (Qiagen) followed by cDNA synthesis of 4 μg RNA using random hexamer primers, and a mix of ribonuclease inhibitors, dNTPs, and GoScript reverse transcriptase (Promega). *PSPH* mRNA expression together with *GAPDH* or *Hprt* as a reference gene, were analyzed in triplicates using SYBR Green-based qRT-PCR on a CFX Connect instrument (Bio-rad laboratories). Relative mRNA expression from four technical repeats was determined using the ΔΔCt method (primer sequences, Supplementary Data 10).

**T-ALL patient samples**. This study was approved by the ethics committees of UZ/KU Leuven and Universiteit Gent (S54608 and S59975). After getting written informed consent, the mononuclear cell fraction of bone marrow from pediatric T-ALL patients was obtained and the cells were xenografted into 6–8 weeks old NSG mice. The public R2: Genomics Analysis and Visualization Platform was used to explore mRNA expression levels in pediatric T-ALL and normal bone marrow CD34+ control samples (dataset: Meijerink, MAS5.0-u133p2, this study includes 117 pediatric T-ALL and 7 normal bone marrow control samples).

**Xenografting in NOD.Cg-Prkdc^scid Il2rg^tm1Wjl/SzJ (NSG) mice**. NSG animal experiments were approved by the KU Leuven animal ethics committee (ECD; approval P179/2015). Human T-ALL patient samples were engrafted by tail vein injections of $10^6$ cells. Blood samples were obtained every 2 weeks from the facial vein. Cells were stained for 30 min with 4 μl human CD45 antibody (eBioscience) and measured by flow cytometry. Animals reaching > 90% human CD45 expressing cells in their peripheral blood were sacrificed. Terminal cardiac bleeding was performed under pentobarbital (150 mg/kg) anesthesia to obtain blood plasma. Human T-ALL patient-derived xenograft (PDX) X12, KE37, and RPMI8402 cells were transduced with scrambled or shRNAs targeting PSPH (shPSPH) and engrafted by tail vein injection of 1*10$^6$ cells.

**Statistics**. All statistical analyses were performed using R or IBM SPSS 23 (IBM Analytics) softwares. A two-sided Wald test was applied by DESeq2 to analyze differentially expressed transcripts. *p*-values were corrected for multiple comparisons by the Benjamini–Hochberg method. We used the two-sided *z*-test in to analyze differences in TE. Obtained *p*-values were corrected for multiple comparisons by the Benjamini–Hochberg method. A *t*-test was used by Scaffold4 for the analysis of differentially expressed proteins, as previously described[11]. Enrichment analyses performed with WebGestalt were based on a hypergeometric test with Benjamini–Hochberg correction.

A two-tailed Student's *t*-test was used to compare *RPL10 WT* and *RPL10 R98S* conditions for all in vitro and in vivo measurements based upon Levene's Test for Equality of Variances. For the comparisons between scrambled controls and shPSPH cells, Student's *t*-tests were used to compare groups individually. The proliferation index was calculated by taking the mean out of at least three measurements for the fold changes in growth comparing scrambled controls and

shPSPH cell cultures. The boxplots present five sample statistics; the minimum, the lower quartile, the median, the upper quartile, and the maximum, all summarized in a visual display using IBM SPSS software. Whiskers/inner fences are defined based upon the distribution of the data points, up to a maximum of 1.5 times the height of the box. Outliers are presented in each plot, when exceeding the whiskers. Rounded outliers lay in the range of 1.5–3 times the height of the box and asterisks or stars are extreme outliers that have values more than three times the height of the boxes.

**Reporting summary**. Further information on research design is available in the Nature Research Reporting Summary linked to this article.

## Data availability

The quantitative mass spectrometry dataset has been described previously (PRIDE identifier PXD005995)[11], as well as polysomal RNA sequencing and its associated mRNA sequencing (GEO accession GSE106528)[11] and a supplementary mRNA sequencing dataset (GEO accession GSE106528)[12]. Ribosome footprinting and matched mRNA sequencing (GEO accession GSE106528) have not been described previously. All datasets are listed in Supplementary Data 11. Number of reads and reproducibility of the high-throughput sequencing libraries are also provided in Supplementary Table 11. These datasets are linked to Figs. 1, 2 and 3 and Supplementary Figs. 1–4. There are no restrictions on data availability.

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

## Acknowledgements

We thank Daniele Pepe for analyzing glycine and serine contents in the proteins detected by mass spectrometry. This research was funded by an ERC starting grant (334946), FWO funding (G067015N, G084013N, and 1509814N), by grants from Stichting Tegen Kanker (2012-176, 2016-775, and 2016-801) and by funding from the KU Leuven Research Council (C1 grant C14/18/104) to K.D.K. K.R.K. was supported by the Lady Tata Memorial Trust International Award for research in Leukemia and by a leukemia research grant from the "Me To You" Foundation. T.G. was supported by a fellowship "Emmanuel van der Schueren" from Kom op tegen Kanker. S.O.S. was supported by a Jose Carreras EHA junior research grant. B.V. and S.V. are SB Ph.D. fellow at FWO (nos. 1S07118N and 1S49817N). G.R. is supported by consecutive fellowships from "Emmanuel van der Schueren" from Kom op tegen Kanker and from FWO. S.-M.F. acknowledges funding from FWO and KU Leuven Methusalem Co-funding. P.V. and D.C. have senior clinical investigator fellowships of the FWO.

## Author contributions

K.R.K. designed and performed research, analyzed data, and wrote the manuscript. L.F. designed and performed analyses of all omics datasets and wrote the manuscript. T.G. generated the omics datasets. G.R., M.P., and S.-M.F. performed and interpreted the $^{13}C_6$-Glucose GC–MS analyses. S.O.S. generated graphics and edited the manuscript. F.L.-P. and R.A. generated ribosome footprinting libraries. B.V. and S.V. generated the CRISPR-Cas9 Jurkat clones. J.V., J.O.d.B. and J.R. provided technical experimental support. P.V. and D.C. measured metabolites in serum samples. J.C. provided Cas9 reagents and primary T-ALL xenograft samples. M.F. supervised bioinformatics studies. K.D.K. designed research, supervised the study, and wrote the manuscript.

## Additional information

**Competing interests:** The authors declare no competing interests.

