## [Peer Review File · Nature Communications]

Reviewers' comments:

Reviewer #1 (Remarks to the Author):

In this manuscript Kampen and colleagues apply an integrated translational/proteomic approach to identify genes specifically deregulated by the T-ALL associated RPL10 R98S mutation. They find a significant upregulation of PSPH, involving serine and glycine metabolism.

This result is novel and of general interest. The manuscript is well written and the approach described is comprehensive and complete.

My major concern is that no attempt is made to characterise the metabolic effects of increased levels of serine/glycine in leukaemic cells. Do cells take advantage of increased serine/glycine levels because they use it for protein synthesis? In this sense this information could be easy to obtain and would complement very well what is shown in Figure 5 and 6: Does the addition of serine/glycine increase protein synthesis in RPL10 Wt cell but not in mutant cells? If this is not the case one should consider different possible metabolic pathways involving these amino-acids (e.g. nucleic acids, lipids).

These results could be taken into consideration also in the discussion section and in Figure 8.

Minor Points

Labeling (A to F) in figure 4 should be revised.

Reviewer #2 (Remarks to the Author):

In this manuscript Kampen and colleagues focus on RPL10, a ribosomal protein, mutated in paediatric T-cell acute lymphoblastic leukemia (T-ALL) and, in particular, on the R98S mutant, because this mutation is reported in almost all patients where RPL10 is mutated.

In a previous publication they compared the isogenic cell lines Ba/F3 wt and RPL10 R98S by MS and have identified that 4% of the proteins they could detect by MS were differentially expressed. In the current manuscript, looking for a mechanism, they perform mRNA sequencing and ribosome footprinting. Analysis of these data via a set of approaches leads them to identify (i) the transcription factor Ikzf2/Helios as overexpressed in the cell line carrying the mutation and (ii) PSPH as the only protein whose level of expression changes consistently with changes in translational efficiency. Because PSPH is an enzyme involved in serine biosynthesis, the authors then move on testing in a series of in vitro and in vivo models the importance of serine biosynthesis in T-ALL and conclude that PSPH could be a novel therapeutic target in T-ALL.

The paper is well written and the topic is interesting. Indeed, serine biosynthesis has been shown to be important in a series of solid tumours, thus adding T-ALL to the list would confirm the crucial role of this metabolic pathway in cancer.

However, while the omics approach to explain the effect of RPL10 R98S is appreciated, the data shown in this manuscript to demonstrate that this mutation regulates serine biosynthesis are not convincing.

Major points:

1. Fig 4 C: the WB for PSPH shows no difference between the two cell lines (Jurkat RPL10 wt vs RPL10 R98S), while tubulin, used as loading control, is changing. Despite this, the authors quantify the WB and conclude PSPH is overexpressed. This is not convincing, and the authors should aim to have equal loading between the different lanes and show differences in PSPH.
2. Fig 4D: bone marrow cells derived from Rpl10 R98S knock-in show upregulation of PSPH only after second re-plating. Could the authors provide an explanation for this?
3. Fig 4E: if serine is produced from U13C-glucose, it should be detected as M+3, considering that

3PG is the initial metabolite used in the serine biosynthesis pathway. The data are showing either an increase of m+1 and m+2 in the Ba/F3 cells or no increase at all in the labelled serine in the Jurkat cells. These data do not support serine biosynthesis from glucose, and the authors do not comment on where the m+1 and m+2 isotopologues are coming from. Glycine instead is shown as labelled m+2, which might come from glucose, and maybe this should be the focus? However an explanation of how PSPH overexpression leads to glycine biosynthesis is not clear. The authors indicate that possibly also SHMT2 is overexpressed but more data would be required to demonstrate this and more attention should be given to SHMT2 if indeed glycine biosynthesis would become more relevant in the manuscript.

4. Are other amino acids or metabolites changing when comparing the cells overexpressing wt vs R98S RPL10 mutants? Only serine and glycine are shown, thus the specificity of the effect is unclear.

5. Fig S10 shows the level of expression of PSPH mRNA in a set of T-ALL cell lines and patient samples. In most of the cases the data show elevated PSPH mRNA expression, but for example, as the authors state, only two of the patients carry the RPL10 R98S mutation. How can this be explained? These data would suggest that PSPH is upregulated in most of T-ALL patients independently of RPL10 R98S mutation, and while the data are interesting, they also raise critical questions about the link between RPL10 R98S mutation and PSPH regulation.

6. Fig 5A compares the level of circulating metabolites in mice xenografted with T-ALL samples vs healthy control mice. Were the xenografts harbouring the mutation? Also, a xenograft that is not supposed to induce serine/glycine secretion is missing as important control that the observed phenotype is specific.

7. Fig 5B shows that a higher concentration of serine and glycine in the medium where Ba/F3 RPL10 R98S mutant cells grew in comparison to wt cells. What is the proliferation rate of these two cell lines? Is this effect specific for serine/glycine or are also other metabolites left in higher concentration possibly due to a different proliferation rate? The same question applies to Fig 5C. Can any other metabolite than serine rescue the medium conditioned by wt cells?

8. The WB in Fig 6C is not sufficient to claim an effect on cell cycle. A proper cell cycle profile would be required. This is an important point and need further validation.

To conclude, while there seems to be a role for serine biosynthesis in T-ALL, the data linking the RPL10 R98S mutation to PSPH regulation are not convincing.

On a different note, the data about Ikzf2 (Helios) are also interesting, but there is no follow up. Are genes regulated by Ikzf2 (Helios) differentially expressed? What is the function of induced expression of Ikzf2 (Helios) in cells harbouring RPL10 R98S?

Reviewer #3 (Remarks to the Author):

RPL10 mutant T-ALL is characterised in this manuscript, using a multi-omics approach. This work builds on a recent Leukemia paper (RNA seq and protein expression) and adding translational analysis through ribosomal footprinting (RPFseq) analysis.

The initial 3 figures concentrate on this work, and are largely correlative, demonstrating that gene regulation in this model is primarily controlled through transcriptional expression, with a minority contribution through translation. There was surprisingly little overlap between the 2. This initial work is novel for the use of RPFseq, however is largely incremental on previous publications. The work is largely dependent on the isogenic BAF3 lines generated in previous work.

A novel target, Psph, was identified and validated using protein quantification and metabolic analysis in the transgenic BAF3 and Jurkat cell lines. Increased RNA expression is noted in a number of cell lines and human samples (S10) vs. a single control sample (thymus), and validated with public datasets (4F).

Primary PDX were generated and metabolic analysis revealed dysregulation compared to healthy

controls (presumably non-engrafted NSG mice). metabolic supplementation had in vitro effects on BAF3 RPL10WT cells, not RPLR98S cells, and also on BM stromal and myeloid cells genetic knockdown of PSPH was detrimental to ALL cell lines in vitro, with increased apoptosis and also in xenograft experiments. This work is logical and correlates well between in vitro cell line work and the phenotype observed in vivo.

Comments:

The levels of serine and glycine fluctuate markedly in the plasma over time (Fig 5A) and the reasons for this are unclear, and not discussed in detail.

Metabolic supplementation was not performed in vivo, or using primary human ALL cells.

The PDX studies appear to be compared with NSG mice, rather than NSG mice engrafted with human CD34 cells, which would be a more appropriate control.

The PDX studies were not used to validate therapeutic targeting of PSPH in this work, rather cell line work was used throughout.

Reviewer #1 (Remarks to the Author):

In this manuscript Kampen and colleagues apply an integrated translatic/proteomic approach to identify genes specifically deregulated by the T-ALL associated RPL10 R98S mutation.

They find a significant upregulation of PSPH, involving serine and glycine metabolism.

These result is novel and of general interest. The manuscript is well written and the approach described is comprehensive and complete.

My major concern is that no attempt is made to characterize the metabolic effects of increased levels of serine/glycine in leukaemic cells. Do cells take advantage of increased serine/glycine levels because they use it for protein synthesis? In this sense this information could be easy to obtain and would complement very well what is shown in Figure 5 and 6: Does the addition of serine/glycine increase protein synthesis in RPL10 Wt cell but not in mutant cells? If this is not the case one should consider different possible metabolic pathways involving these amino-acids (e.g. nucleic acids, lipids).

These results could be taken into consideration also in the discussion section and in Figure 8.

Answer: We agree with the reviewer that it is of interest to define for what purpose the leukemic cells use the increased levels of serine/glycine. As highlighted by the reviewer, cells can take advantage of increased serine/glycine for protein synthesis. To test this hypothesis, we performed O-propargyl-puromycin (OPP) protein synthesis flow cytometry analysis of Rpl10 WT and R98S mouse bone marrow cells and of PSPH knockdown T-ALL cells (KE37, RPMI8402, DND41, and T-ALL PDX X12). This revealed that *de novo* protein translation was not changed in Rpl10 R98S cells and not affected in PSPH knockdown cells as compared to scrambled control T-ALL cells. This is now included in Figures 4F and 6F and in text lines 293-294 and 359-360.

Figure legend: *Flow cytometry analysis of de novo protein translation by O-propargyl-puromycin (OPP) incorporation. Left: lineage negative bone marrow cells from Rpl10 WT and R98S mice (Results from triplicate samples from two independent mouse donors are shown and relative mean*

fluorescent intensity (MFI) is plotted.). Right; PSPH knockdown T-ALL cells. Relative protein translation as shown in the figure was calculated as shPSPH#1 and shPSPH#2 OPP MFI relative to scrambled control cells and results for KE37, RPMI8402 and DND41 were combined. All box-plots show the median and error bars define data distribution.

Another possibility is that serine catabolism would be used to generate formate, which can then be incorporated into purines to facilitate nucleotide synthesis (as shown now in Figure 4A). Since this mechanism has been described in activated T-cells and oxidative cancers (Nat Commun. 2018 Apr 10;9(1):1368; Sci Adv. 2016 Oct 28;2(10):e1601273; Cell Metab. 2017 Feb 7;25(2):345-357), we looked into the effects of RPL10 R98S and PSPH knockdown on the generation of formate. Our analysis revealed that RPL10 R98S mutant lin- mouse bone marrow cells present ~40% higher formate levels as compared to RPL10 WT lin- mouse bone marrow cells. In our PSPH knockdown T-ALL cell models, formate levels were decreased by 20-40% when PSPH protein levels were downregulated. **These observations suggest that the enhanced serine/glycine turnover mainly functions to fuel formate generation, which can serve for purine synthesis.** These findings are now included in Figures 4G and 6E and in text lines 294-297, 356-361 and 439-443 and Figure 8 was adapted accordingly as requested.

Figure legend: **Formate analysis.** Left: Relative formate derived NAD(P)H levels in the bone marrow of WT and R98S mutant mice. Background NAD(P)H levels were subtracted from formate derived NAD(P)H levels and all samples were corrected for total protein input. The box-plots include combined results of two independent experiments comparing three WT versus three R98S mutant BM CFC assay samples derived from independent donor mice. Right: Relative formate derived NAD(P)H levels in scrambled control and PSPH knockdown T-ALL cells (combined results for KE37, RPMI8402, DND41, X12). Background NAD(P)H levels were subtracted from formate derived NAD(P)H levels and corrected for total protein input. The box-plots include combined results of two independent experiments.

Minor Points

Labeling (A to F) in figure 4 should be revised. We fixed this in the manuscript.

Reviewer #2 (Remarks to the Author):

In this manuscript Kampen and colleagues focus on RPL10, a ribosomal protein, mutated in paediatric T-cell acute lymphoblastic leukemia (T-ALL) and, in particular, on the R98S mutant, because this mutation is reported in almost all patients where RPL10 is mutated.

In a previous publication they compared the isogenic cell lines Ba/F3 wt and RPL10 R98S by MS and have identified that 4% of the proteins they could detect by MS were differentially expressed. In the current manuscript, looking for a mechanism, they perform mRNA sequencing and ribosome footprinting. Analysis of these data via a set of approaches leads them to identify (i) the transcription factor Ikzf2/Helios as overexpressed in the cell line carrying the mutation and (ii) PSPH as the only protein whose level of expression changes consistently with changes in translational efficiency. Because PSPH is an enzyme involved in serine biosynthesis, the authors then move on testing in a series of in vitro and in vivo models the importance of serine biosynthesis in T-ALL and conclude that PSPH could be a novel therapeutic target in T-ALL.

The paper is well written and the topic is interesting. Indeed, serine biosynthesis has been shown to be important in a series of solid tumours, thus **adding T-ALL to the list would confirm the crucial role of this metabolic pathway in cancer.**

However, while the omics approach to explain the effect of RPL10 R98S is appreciated, the data shown in this manuscript to demonstrate that this mutation regulates serine biosynthesis are not convincing.

Answer: We show **increased PSPH transcription in RPL10 R98S expressing Ba/F3 cells by mRNA sequencing (3 datasets) and qRT-PCR confirmation** and additionally by qRT-PCR confirmation in Jurkat RPL10 WT and R98S clones. Ribosome footprinting of Ba/F3 RPL10 WT and R98S clones revealed that on top of 5-fold increased *Psph* transcription, R98S cells display a **2-fold increase in PSPH translation efficiency**. To further confirm this elevation in translation efficiency, we now added polysome profiling by sucrose gradient fractionation followed by qRT-PCR analysis of ribosome bound mRNA. We observed a strong **shift of the distribution of ribosome bound PSPH mRNA towards the most actively translated polysomal fractions in RPL10 R98S cells as compared to WT cells** (now shown in Figure S9 and in text lines 271-274). These data thus collectively support increase in overall PSPH translation by induced mRNA expression as well as translational efficiency. Finally, we show that the increased PSPH protein expression levels enforce a higher glucose to serine/glycine conversion, supported by **elevated levels of labeled carbons derived from $^{13}\text{C}_6$ -glucose**, which emphasizes **increased *de novo* serine and glycine synthesis in both Ba/F3 clones and Jurkat RPL10 WT and R98S clones**. We believe that this thorough validation of the effects of introducing the RPL10 R98S mutation into different lymphoid and leukemic cell models is strongly supporting the contribution of the R98S mutation to enhance serine/glycine metabolism in T-ALL.

Figure legend: Polysomal qRT-PCR analysis of the PspH mRNA in RPL10 WT versus R98S Ba/F3 cells. Cell lysates from 3 different RPL10 WT Ba/F3 cell clones (WT#28, WT#29 and WT#36) and three RPL10 R98S Ba/F3 cell clones (R98S#11, R98S#13 and R98S#35) were applied on a sucrose gradient and fractionated into 7 fractions as indicated on the left, heavy polysomal fractions 5-7 were combined. RNA was isolated from each fraction, reverse transcribed, and qRT-PCR was performed for the PspH mRNA. The plot shows the distribution of the PspH mRNA over the different fractions (the % of PspH mRNA in the analyzed fraction is shown, with the PspH mRNA in all fractions of that sample together being 100%) and illustrates a shift of the distribution of ribosome bound PSPH mRNA towards the most actively translated polysomal fractions in RPL10 R98S cells as compared to WT cells.

1. Fig 4 C: the WB for PSPH shows no difference between the two cell lines (Jurkat RPL10 wt vs RPL10 R98S), while tubulin, used as loading control, is changing. Despite this, the authors quantify the WB and conclude PSPH is overexpressed. This is not convincing, and the authors should aim to have equal loading between the different lanes and show differences in PSPH.

Answer: We agree with the reviewer and performed new immunoblot analysis of PSPH in Jurkat RPL10 WT and R98S clones. This is now shown in Figure 4C.

2. Fig 4D: bone marrow cells derived from Rpl10 R98S knock-in mouse show upregulation of PSPH only after second re-plating. Could the authors provide an explanation for this?

Answer: The replating experiment was repeated several times now. In all experiments, upregulation of PSPH was observed in the RPL10 R98S as compared to RPL10 WT expressing bone marrow cells as we show in **Figure 4D** (including statistical analysis now). In some experiments, this phenotype was

already observed at first replating, in others only from the second replating onwards. We therefore removed the comment on the second replating in the manuscript text.

3. Fig 4E: if serine is produced from U13C-glucose, it should be detected as M+3, considering that 3PG is the initial metabolite used in the serine biosynthesis pathway. The data are showing either an increase of m+1 and m+2 in the Ba/F3 cells or no increase at all in the labelled serine in the Jurkat cells. These data do not support serine biosynthesis from glucose, and the authors do not comment on where the m+1 and m+2 isotopologues are coming from. Glycine instead is shown as labelled m+2, which might come from glucose, and maybe this should be the focus? However an explanation of how PSPH overexpression leads to glycine biosynthesis is not clear. The authors indicate that possibly also SHMT2 is overexpressed but more data would be required to demonstrate this and more attention should be given to SHMT2 if indeed glycine biosynthesis would become more relevant in the manuscript.

Answer: We agree that serine directly produced from glucose is detected as M+3 and can be further converted to M+2 glycine. Nevertheless, M+2 glycine can be converted back into serine, as serine-glycine conversion is a reversible reaction. Both M+1 and M+2 serine carbon labeling can occur when conversions between serine and glycine take place. This is now explained in manuscript lines 284-287.

For the $^{13}\text{C}_6$ glucose tracing in the Jurkat model (**Figure 4E**): The PSPH protein overexpression we observed in Jurkat cells is less pronounced compared to the PSPH overexpression the Ba/F3 model (see Figure panels 3B versus 3C), and therefore the serine/glycine synthesis phenotype is less pronounced in Jurkat cells in the metabolic tracer experiment. Nevertheless, as we wrote on text lines 289-291, RPL10 R98S mutant Jurkat cells showed an elevation in total labeled serine (M+1, M+2 and M+3 together) and glycine (M+1 and M+2) contribution from $^{13}\text{C}_6$ -Glucose (**Figure 4E**, total labeled serine $p = 0.020$, total labeled glycine $p = 0.002$).

Our data support high serine/glycine exchange, therefore our M+3 is increased to a lesser extent than M+1/+2. Yet, all labeled carbons that are detected in M+1 and M+2 originate from labeled glucose. It is likely that high serine/glycine turnover supports the needs of our cells for both serine and glycine. SHMT2 was slightly induced in our mass spectrometry data, which supports the observed high serine/glycine exchange (**Figure S10** and text lines 287-289). We agree with the reviewer that our data support that serine is not the endpoint, and as outlined above we now also added data supporting further catabolism of serine into glycine and formate (Figures 4F + 6E) and we explain this point in text lines 294-297, 356-361 and 439-443. Accordingly, we changed figure 8 as well as the title and text of our manuscript to emphasize that not only serine but serine/glycine metabolism is altered in our RPL10 R98S (increased) and PSPH knockdown (decreased) models.

4. Are other amino acids or metabolites changing when comparing the cells overexpressing wt vs R98S RPL10 mutants? Only serine and glycine are shown, thus the specificity of the effect is unclear.

Answer: Other metabolites in the conditioned media of R98S clones were not changed (see figure below). We only observed that R98S clones consumed more alanine compared to RPL10 WT clones.

5. Fig S10 shows the level of expression of PSPH mRNA in a set of T-ALL cell lines and patient samples. In most of the cases the data show elevated PSPH mRNA expression, but for example, as the authors state, only two of the patients carry the RPL10 R98S mutation. How can this be explained? These data would suggest that PSHP is upregulated in most of T-ALL patients independently of RPL10 R98S mutation, and while the data are interesting, they also raise critical questions about the link between RPL10 R98S mutation and PSPH regulation.

Answer: We regret that the message we want to convey in our manuscript was not clearly enough explained. We describe that **RPL10 R98S upregulates PSPH**, but that this effect is not restricted to RPL10 R98S mutant cases and that **almost all T-ALL samples show increased expression levels of PSPH, as also the reviewer concludes**. To make this point more clear, we now added extra text on lines 300-301. Moreover, we now included an immunoblot analysis (Figure S11B), which shows that PSPH protein expression levels are induced in RPL10 WT and R98S T-ALL as compared to normal bone marrow CD34 cells and pediatric AML cases. Also in this T-ALL dataset, RPL10 WT T-ALL xenografted samples (X10-X12) express similarly high PSPH protein levels as compared to RPL10 R98S xenografted samples (X13-X15). Our data thus support that the RPL10 R98S mutation in pediatric T-ALL samples can explain elevated PSPH expression in the 10% of cases in which this mutation occurs, while other mechanisms will contribute to PSPH overexpression in RPL10 WT T-ALL cases. As we indicate in discussion lines 445-446, serine synthesis has for example been described to be upregulated in Cyclin D3:CDK4/6 complex-driven T-ALL.

Figure legend: Immunoblot analysis of PSPH levels in CD34 positive normal bone marrow samples (NBM #1 and NBM #2), T-ALL xenografts (X10-X15) and AML samples (AML #1, #2 and #3). Vinculin served as loading control.

6. Fig 5A compares the level of circulating metabolites in mice xenografted with T-ALL samples vs healthy control mice. Were the xenografts harbouring the mutation? Also, a xenograft that is not supposed to induce serine/glycine secretion is missing as important control that the observed phenotype is specific.

Answer: The samples indicated in bold and underscored are the T-ALL samples containing the R98S mutation, the other are RPL10 WT T-ALL xenografts. To make this more clear, we now indicated the RPL10 R98S T-ALL PDX samples in blue, which is more consistent with the other manuscript figures.

The increased serine/glycine levels in the plasma of T-ALL xenografted mice was not exclusive for R98S mutant T-ALL samples, as we observed that all T-ALL xenografted mice presented elevated levels of serine/glycine synthesis pathway metabolites except for X13. As a proper control that shows T-ALL specificity for the induced serine/glycine plasma levels, data from therapy (venetoclax) suppressed T-ALL PDX mice are shown below. In untreated X15 engrafted NSG animals, T-ALL cells were present in the peripheral blood and serine and glycine levels were induced as compared to age-matched control NSG mice. In contrast, in the animals venetoclax therapy controlled X15 engrafted NSG animals lacking circulating T-ALL cells in the peripheral blood, plasma serine and glycine levels were brought back to the level of control non-xenografted NSG mice. These findings make us conclude that our observed phenotypes are specific and originate from circulating T-ALL cells in the blood of the xenograft mice.

*Figure legend: Plasma serine and glycine levels in control NSG mice (ctrl NSG), NSG mice injected with RPL10 R98S mutant xenograft sample X15 analyzed at disease end stage (X15 NSG) and NSG mice injected with xenograft sample X15 and treated with anti-leukemic venetoclax therapy (X15 therapy controlled NSG). ** $p < 0.01$.*

7. Fig 5B shows that a higher concentration of serine and glycine in the medium where Ba/F3 RPL10 R98S mutant cells grew in comparison to wt cells. What is the proliferation rate of these two cell lines? Is this effect specific for serine/glycine or are also other metabolites left in higher concentration possibly due to a different proliferation rate? The same question applies to Fig 5C. Can any other metabolite than serine rescue the medium conditioned by wt cells?

Answer: Other metabolites in the conditioned media of R98S clones were not changed (see figure below). We only observed that R98S clones consumed more alanine compared to RPL10 WT clones, which was not supported by $^{13}\text{C}_6$ -glucose flux analysis in R98S clones.

The conditioned media still contains high leftovers of glycine, alanine, proline etc. Therefore, we do not expect that these metabolites can provide the rescue observed in Fig 5C (now Figure 5D). The lack of serine in RPL10 WT conditioned media provided a rescue opportunity for serine, which was still present in the media of RPL10 R98S clones. This rescue opportunity is not likely for other metabolites, as these were not fully consumed in WT conditioned media.

8. The WB in Fig 6C is not sufficient to claim an effect on cell cycle. A proper cell cycle profile would be required. This is an important point and needs further validation.

Answer: The reduced proliferation in growth curve analysis (proliferation index) and immunoblot analysis of phospho-CDK2 tyr160 indicates cell cycle defects induced by PSPH targeting. As requested, we now further validated the cell cycle phenotype by showing that PSPH knockdown T-ALL cells show a decrease in the percentage of cycling cells as assessed by BrdU incorporation and PI cell cycle analysis. This is now included in Figure 6D and in text lines 355-356.

Figure legend: Left histograms: BrdU incorporation or PI cell cycle flow cytometry analysis of representative scrambled control and PSPH knockdown T-ALL cell lines. Right: Quantification of the percentage cycling cells in cultures of scrambled, shPSPH#1, and shPSPH#2 KE37, DND41, and RPMI8402 T-ALL cells. For technical reasons, some T-ALL lines were only analyzed by either BrdU or PI cell cycle analysis, and at least in two independent experiments per sample.

To conclude, while there seems to be a role for serine biosynthesis in T-ALL, the data linking the RPL10 R98S mutation to PSPH regulation are not convincing.

Answer: As explained above, we applied different techniques (RNA sequencing, qRT-PCR validation, ribosome footprinting, and qRT-PCR analysis of ribosome bound mRNA) and used several models (Ba/F3 isogenic and Jurkat CRISPR/Cas9 models) to show that **introduction of the RPL10 R98S mutation enhances PSPH mRNA transcription and provides the mutant ribosome with a preference for PSPH translation, which functionally leads to elevated serine/glycine synthesis as supported by e.g. ¹³C₆-glucose tracing**. We believe this comprehensive analysis convincingly demonstrates the link between RPL10 R98S and PSPH upregulation.

On a different note, the data about Ikzf2 (Helios) are also interesting, but there is no follow up. Are genes regulated by Ikzf2 (Helios) differentially expressed? What is the function of induced expression of Ikzf2 (Helios) in cells harbouring RPL10 R98S?

Answer: 128 target genes of Ikzf2/Helios are indeed differentially expressed (128 upregulated genes can be found in Table S6), and therefore Helios is picked up as master regulator in the iREGULON analysis. Additionally, we show enhanced Ikzf2/Helios protein expression in our RPL10 R98S models (Figure 2E). In the discussion text lines 396-401, we speculate about the possible role of Ikzf2/Helios in T-ALL and its function in the healthy functioning hematopoietic system.

Reviewer #3 (Remarks to the Author):

RPL10 mutant T-ALL is characterised in this manuscript, using a multi-omics approach. This work builds on a recent Leukemia paper (RNA seq and protein expression) and adding translational analysis through ribosomal footprinting (RPFseq) analysis.

The initial 3 figures concentrate on this work, and are largely correlative, demonstrating that gene regulation in this model is primarily controlled through transcriptional expression, with a minority contribution through translation. There was surprisingly little overlap between the 2. This initial work is novel for the use of RPFseq, however is largely incremental on previous publications. The work is largely dependent on the isogenic BAF3 lines generated in previous work.

Answer: While we agree that the current manuscript certainly builds on previous findings, it provides broad integration of a multi-omics approach with a systematic analysis of the impact of the RPL10 R98S mutation on the cellular transcriptome and translatoome. As far as we are aware, this is the first time such a complete analysis of a disease linked ribosomal defect is performed.

A novel target, PspH, was identified and validated using protein quantification and metabolic analysis in the transgenic BAF3 and jurkat cell lines. Increased RNA expression is noted in a number of cell lines and human samples (S10) vs. a single control sample (thymus), and validated with public datasets (4F).

Answer: We now added extra data showing that PSPH protein expression is strongly elevated in T-ALL PDX samples as compared to human CD34 positive normal bone marrow cells and acute myeloid leukemia samples. This was included in Figure S11B.

Figure legend: Immunoblot analysis of PSPH levels in CD34 positive normal bone marrow samples (NBM #1 and NBM #2), T-ALL xenografts (X10-X15) and AML samples (AML #1, #2 and #3). Vinculin served as loading control.

Primary PDX were generated and metabolic analysis revealed dysregulation compared to healthy controls (presumably non-engrafted NSG mice).

Metabolic supplementation had in vitro effects on BAF3 RPL10WT cells, not RPLR98S cells, and also on BM stromal and myeloid cells.

Genetic knockdown of PSPH was detrimental to ALL cell lines in vitro, with increased apoptosis and also in xenograft experiments. This work is logical and correlates well between in vitro cell line work and the phenotype observed in vivo.

Comments:

The levels of serine and glycine fluctuate markedly in the plasma over time (Fig 5A) and the reasons for this are unclear, and not discussed in detail.

Answer: It seems that Figure 5A (now Figure 5B) has been misinterpreted, as we analyzed serine and glycine plasma serum levels only at one time point: the T-ALL leukemia disease end stage at which all animals present >80% human CD45 positive T-ALL cells in their peripheral blood (hCD45 analysis). We are not able to measure serine/glycine plasma levels over time, due to the limited amount of blood that we are allowed to collect when the animals are alive. The serine/glycine levels do not seem to fluctuate much, as these are mean plasma levels of 4-5 independent mice per xenograft. For each individual T-ALL PDX sample, we therefore observe low variance in the serine/glycine plasma levels. Like in patients, we do observe differences between different T-ALL PDX samples, which is expected due to the high mutational heterogeneity in leukemic patient samples.

Metabolic supplementation was not performed *in vivo*, or using primary human ALL cells.

Answer: The supplementation experiments in our manuscript support that non-leukemic niche cells can benefit from the serine/glycine excreted from T-ALL cells. The *in vivo/ex vivo* effects of metabolic supplementation will therefore not affect leukemic cells, as these cells provide themselves with sufficient serine/glycine via *de novo* synthesis. To address the *in vivo* dynamics of serine/glycine release by T-ALL cells, we were able to show that therapeutic suppression of leukemic cells into the circulation also blocks their ability to secrete serine/glycine into the circulation.

Figure legend: Plasma serine and glycine levels in control NSG mice (ctrl NSG), NSG mice injected with RPL10 R98S mutant xenograft sample X15 analyzed at disease end stage (X15 NSG) and NSG mice injected with xenograft sample X15 and treated with anti-leukemic venetoclax therapy (X15 therapy controlled NSG). ** $p < 0.01$.

The PDX studies appear to be compared with NSG mice, rather than NSG mice engrafted with human CD34 cells, which would be a more appropriate control.

Answer: We believe that the therapy suppressed T-ALL xenograft model is a better control compared to normal human CD34 cells, because normal human CD34 cells do not engraft or expand in the bone marrow of NSG mice. Upon *in vivo* venetoclax treatment, these mice show human T-ALL engraftment in the bone marrow but complete repression of leukemic cells to the circulation. Circulating

serine/glycine levels were brought back to the level of control non-xenografted NSG mice (see above). Therefore, we concluded that the non-xenografted NSG mice serve as a proper control to measure basal serine/glycine levels in the circulation of NSG mice.

The PDX studies were not used to validate therapeutic targeting of PSPH in this work, rather cell line work was used throughout.

Answer: We agree that it would be interesting to target PSPH in a PDX T-ALL mouse model. As there are no PSPH targeting compounds available yet, we tried shRNA targeting of PSPH in the PDX X12 T-ALL sample, which presents the highest PSPH protein expression levels in an *in vivo* xenograft setting. Scrambled control, shPSPH1 and shPSPH2 transduced PDX X12 T-ALL cells were injected in five mice per group. Transduction of X12 T-ALL cells was highly efficient (Figure panel A). Unfortunately, the transduction of primary T-ALL PDX cells probably caused an *in vivo* expansion disadvantage, even in the scrambled control condition. This leads to the accumulation of non-transduced X12 T-ALL cells in these PDX mice, giving rise to leukemia as evidenced by up to 75% of CD45 positive leukemia cells in the peripheral blood and by elevated spleen weights (Figure panel C-left and D-left). Nevertheless, mCherry positive scrambled control X12 T-ALL cells were detected at low percentage (1-6%) *in vivo* in all NSG mice at all leukemic sites (e.g. blood, bone marrow, and spleen) supporting a certain degree of expansion, while all of the shPSPH#1 and shPSPH#2 X12 T-ALL cells were completely lost in NSG mice at all leukemic sites. This finding is in line with our *in vivo* T-ALL cell line data and further supports that targeting of PSPH is repressing primary PDX T-ALL progression *in vivo*. In line with our *in vivo* T-ALL cell line data, the shPSPH#2 with the strongest PSPH knockdown showed the most pronounced *in vivo* effects by reducing the leukemia burden in the blood in three out of the five X12 T-ALL xenografted NSG mice. These findings are now included in the supplementary data Figure S14 and in the text lines 374-376.

Figure S14

Figure legend: PSPH knockdown suppresses in vivo leukemia expansion of X12 PDX T-ALL. A. Flow cytometry analysis of transduction efficiencies of X12 PDX cells. Transduced cells are mCherry positive

B. Left: scheme of experimental design to test the effect of PSPH knockdown on X12 PDX T-ALL leukemia progression in vivo in mice. Middle: immunoblot analysis of PSPH protein expression levels in the X12 PDX T-ALL cells that were injected into NSG recipient mice. Right: quantification of immunoblots shown in the middle. **C.** Left: percentages of human CD45 X12 PDX T-ALL expressing cells in the blood of injected NSG mice two months after tail vein injection. Right: percentage of mCherry expressing cells detected in the peripheral blood of mice transplanted with X12 PDX T-ALL cells. **D.** Left: spleen weights of mice injected with X12 PDX T-ALL cells. Middle: The percentage of mCherry expressing cells detected in the spleens and bone marrow (BM) of mice transplanted with X12 PDX T-ALL shPSPH#1 and shPSPH#2 leukemic cells as compared to scrambled control cell injected mice. Right: reduced leukemia cell invasion in the bone marrow (BM) derived from X12 PDX T-ALL shPSPH#1 and shPSPH#2 leukemic cells as compared to scrambled control cells injected in mice shown by flow cytometric dot-plot analysis. Box-plots show the median and error bars define data distribution. Statistical analysis * p-value < 0.05, ** p-value < 0.01, *** p-value < 0.001.

Reviewers' comments:

Reviewer #1 (Remarks to the Author):

The Authors responded to my concerns and the manuscript improved with revision. I have only one minor comment since in the newly added text (manuscript pages 11, 24 + figure 6 labels and legend (page 28) they mention "protein translation" that should be corrected in "protein synthesis" (or alternatively mRNA translation, which in this case seems less correct).

Reviewer #3 (Remarks to the Author):

The manuscript is clearly written and the presentation of data are clear throughout. The authors provide substantial additional data to address comments raised in the first round of review, specifically addressing i) mechanistic data relating to the metabolic changes within leukemia cells, ii) better in vivo validation of the work including plasma serine/ glycine levels in NSG mice, iii) PDX data supporting the increased expression of PSPH in human disease . Overall the work is substantially improved. I have no further comments or concerns

Reviewer #4 (Remarks to the Author):

In the manuscript "Translatome analysis reveals altered serine/glycine metabolism in T-cell acute lymphoblastic leukemia" Kampen and colleagues used ribosome footprinting to show that an enzyme of serine biosynthesis, PSPH, is upregulated in lymphoid cells harbouring mutations in a ribosomal protein RPL10, usually found mutated in AML patients. They go on and show that serine synthesis is increased in these cells and that this pathway is required for AML cell proliferation.

Overall, this work is interesting and the results on the expression of PSPH upon mutations of RPL10 clear. However, the metabolic characterisation is suboptimal and does not fully support the conclusion that RPL10 mutant cells exhibit increased de novo serine/glycine synthesis. For instance, the increase in glucose-derived serine (m+3) is minimal and not significant in both cell lines investigated (Ba/F3 and Jurkat, Fig. 4E). The only tangible increase in labelling is in m+1 and m+2 serine, indicating that the most likely reaction occurring in these cells is not a conversion from serine to glycine but rather glycine to serine, which might indicate broader dysregulation of the pathway. So, the data in support of activated serine synthesis is scant. Furthermore, the authors would need to provide the expression (mRNA and protein) of the other enzymes of the pathway, not only SHMT2. Finally, the total pool levels of serine and glycine in the cell lines tested is required for completeness.

The argument that cells consume less glycine and serine based on the experiment in Fig. 5C is flawed. Indeed, it could well be that mutant RPL10 cells produce and secrete more serine and glycine. This conundrum could be addressed by measuring the release of labelled glycine and serine upon incubation with ¹³C- glucose, to distinguish it from unlabelled serine in the media. Similarly, the argument that leftover serine in the conditioned media of mutant RPL10 increases cell growth (Fig. 4D) should not be used to conclude that it is serine is causal for the survival benefit. Indeed, another interpretation of this experiment, if the effect of serine is dose-dependent, is that the additional serine does not increase cell proliferation and that other factors present in the conditioned media provided the growth advantage.

Finally, the argument that RPL10 mutant cells use serine and glycine for nucleotide biosynthesis is weak and indirect, since it is based on the determination of formate, which is not a direct readout of nucleotide synthesis. Indeed, as shown by the Vazquez lab (Meiser et al 2018), most cancer cells exhibit a formate overflow i.e. formate production and secretion, as a way to regenerate reducing equivalents in the mitochondria, not directly for nucleotide biosynthesis. Given that

glucose and glycine carbons are also incorporate in nucleotides, the authors should perform a proper assessment of glucose-derived nucleotides to corroborate the authors' conclusions.

Reviewers' comments:

Reviewer #1 (Remarks to the Author):

The Authors responded to my concerns and the manuscript improved with revision. I have only one minor comment since in the newly added text (manuscript pages 11, 24 + figure 6 labels and legend (page 28) they mention "protein translation" that should be corrected in "protein synthesis" (or alternatively mRNA translation, which in this case seems less correct).

We corrected the mistakes in the text and in Figures 4 and 6 and their legends.

Reviewer #2:

He/she questions the clinical relevance of our findings based on Supplementary Figure 11. This reviewer was questioning why PSPH levels were increased in samples both with and without mutant RPL10.

We believe that this comment is based on a misunderstanding. Yes, RPL10 R98S is one of the mechanisms that drives PSPH-mediated serine/glycine synthesis into formate and purines. However, what Supplementary Figure 11 shows is that the majority of T-ALL patients show high PSPH mRNA and protein expression in comparison to normal tissue counterparts such as thymus and bone marrow. This supports that unknown mechanisms besides RPL10 R98S contribute to high PSPH expression in T-ALL. In fact, this finding does not limit but enlarges the clinical relevance of our findings, as this indicates that not only RPL10 R98S positive T-ALL patients will benefit from therapeutic targeting of PSPH. This is also in agreement with the data we show in Figures 6, 7 and Supplementary Figures 14 + 15, where we show that T-ALL cell lines and patient derived xenografts that do not carry the RPL10 R98S mutation but that also express high levels of PSPH display impaired proliferation and *in vivo* expansion upon PSPH downregulation. In this paper, we are the first to show common upregulation of serine/ glycine metabolism in T-ALL and that RPL10 R98S can control this metabolic switch. Moreover, we show that PSPH targeting in T-ALL in general is a novel therapeutic approach, as these leukemic cells seem highly dependent on serine/glycine synthesis for cell proliferation purposes.

Reviewer #3 (Remarks to the Author):

The manuscript is clearly written and the presentation of data are clear throughout. The authors provide substantial additional data to address comments raised in the first round of review, specifically addressing i) mechanistic data relating to the metabolic changes within leukemia cells, ii) better *in vivo* validation of the work including plasma serine/ glycine levels in NSG mice, iii) PDX data supporting the increased expression of PSPH in human disease .

Overall the work is substantially improved. I have no further comments or concerns

Reviewer #4 (Remarks to the Author):

Overall, this work is interesting and the results on the expression of PSPH upon mutations of RPL10 clear. However, the metabolic characterization is suboptimal and does not fully support the conclusion that RPL10 mutant cells exhibit increased *de novo* serine/glycine synthesis. For instance, the increase in glucose-derived serine (m+3) is minimal and not significant in both cell lines investigated (Ba/F3 and Jurkat, Fig. 4E). The only tangible increase in labeling is in m+1 and m+2 serine, indicating that the most likely reaction occurring in these cells is not a conversion from serine to glycine but rather glycine to serine, which might indicate broader dysregulation of the pathway. So, the data in support of activated serine synthesis is scant. Furthermore, the authors would need to provide the expression (mRNA and protein) of the other enzymes of the pathway, not only SHMT2. Finally, the total pool levels of serine and glycine in the cell lines tested is required for completeness.

We have performed an additional tracing experiment and have added data from 5 newly generated Ba/F3 *RPL10 WT* clones and 5 Ba/F3 *RPL10 R98S* clones. We have performed a deeper mass spectrometry analysis of ¹³C₆-Glucose tracing. As we increased the number of Ba/F3 clones, we enhanced our statistical power in this experimental set-up. We do see that serine M+2 and glucose-derived M+3 are both significantly increased in *RPL10 R98S* cells (Figure 4F). We agree that the fact that we also see serine M+2 increasing suggests that there is not only more production of serine from glucose, but also conversion from glycine, as we indicate on lines 215-217 of our revised text. Furthermore, we checked the mRNA and protein levels of all other metabolic enzymes of the serine/glycine pathway and can confirm that besides PSPH, none of the enzymes shows a consistent increase in their mRNA or protein expression in *RPL10 R98S* cells as compared to *RPL10 WT* Ba/F3 cells (Supplementary Table 8 and Supplementary Figure 8; text lines 217-219). This means that only PSPH is increased both transcriptionally and translationally to drive the elevated serine/glycine synthesis and catabolism observed in *RPL10 R98S* cells. Finally, as requested, we have also added the total intracellular serine and glycine pools of our Ba/F3 cell model (Figure 4E; text lines 210-212). This shows significantly elevated serine and glycine levels in the *RPL10 R98S* cells, again supporting more *de novo* serine/glycine synthesis.

The Jurkat cell model indeed shows less pronounced effects on glucose-derived serine as compared to the Ba/F3 cell model. This is however not surprising, considering that this cell line has high baseline PSPH expression levels (Figure 4C- *RPL10 WT* condition) and high baseline m+3 serine labeling (Supplementary Figure 9B - *RPL10 WT* condition). This is in line with our finding that all T-ALL cell lines (which are all *RPL10 WT*) have high PSPH expression (Supplementary Figure 11) and we propose in this work that this is caused by other mechanisms in T-ALL besides *RPL10 R98S* that can drive elevated PSPH expression. In such a background with high baseline PSPH expression and serine/glycine synthesis, it is harder to see the additional effect of *RPL10 R98S*. We rephrased the text to reflect this limitation of the Jurkat cell model (line 219). Nevertheless, the Jurkat model is a nice addition as it is a T-ALL cell line and it does show significant increase in total labeled serine (M+1, M+2 and M+3 together), and in M+1 glycine and in M+2 glycine upon expression of the *RPL10 R98S* mutation (Supplementary Figure 9A-B).

The argument that cells consume less glycine and serine based on the experiment in Fig. 5C is flawed. Indeed, it could well be that mutant *RPL10* cells produce and secrete more serine and glycine. This

conundrum could be addressed by measuring the release of labeled glycine and serine upon incubation with ^{13}C -glucose, to distinguish it from unlabeled serine in the media.

We thank the reviewer for this very constructive comment. As suggested, we have analyzed release of labeled glycine and serine in the conditioned medium upon $^{13}\text{C}_6$ -Glucose tracing of the cells. These results indeed support that the RPL10 R98S cells release more labeled serine/glycine in the environment (Figure 5D). We have also compared general serine and glycine uptake rates of RPL10 R98S versus RPL10 WT cells. These analyses showed that RPL10 R98S cells still take up serine/glycine from the media, but that there is a significantly reduced glycine uptake and a tendency towards less serine uptake as compared to RPL10 WT cells (Figure 5E). Overall, these new results thus suggest that the elevated levels of serine and glycine in the conditioned medium of the RPL10 R98s mutant thus originate from more production and secretion of serine and glycine, with a potential minor contribution from reduced serine/glycine uptake. These new analysis are now included in Figure 5D-E and in text lines 259-261 and 268-271.

Similarly, the argument that leftover serine in the conditioned media of mutant RPL10 increases cell growth (Fig. 5D) should not be used to conclude that it is serine is causal for the survival benefit. Indeed, another interpretation of this experiment, if the effect of serine is dose-dependent, is that the additional serine does not increase cell proliferation and that other factors present in the conditioned media provided the growth advantage.

We believe the reviewer might have misinterpreted Figure 5D. In Figure 5D (now Figure 5F in our revised version), we show that addition of 20 μM serine to conditioned media from RPL10 WT cells can stimulate Ba/F3 cells proliferation as observed for conditioned media from RPL10 R98S cells. This concentration of 20 μM serine exactly corresponds to the excess of serine in RPL10 R98S conditioned medium as compared to RPL10 WT conditioned medium (Figure 5C). The fact that addition of serine to RPL10 WT conditioned medium is sufficient to reproduce the effect of RPL10 R98S conditioned medium supports the causal role of serine in the conditioned medium to support the growth advantage.

Finally, the argument that RPL10 mutant cells use serine and glycine for nucleotide biosynthesis is weak and indirect, since it is based on the determination of formate, which is not a direct readout of nucleotide synthesis. Indeed, as shown by the Vazquez lab (Meiser et al 2018), most cancer cells exhibit a formate overflow i.e. formate production and secretion, as a way to regenerate reducing equivalents in the mitochondria, not directly for nucleotide biosynthesis. Given that glucose and glycine carbons are also incorporate in nucleotides, the authors should perform a proper assessment of glucose-derived nucleotides to corroborate the authors' conclusions.

We performed the $^{13}\text{C}_6$ -Glucose tracing with an increased number of Ba/F3 RPL10 WT and R98S clones to identify whether the R98S clones enhance their serine/glycine synthesis for the production of purines. As such, we added new data showing that the Ba/F3 RPL10 R98S clones have significantly increased M+6, and a tendency of elevated M+7, M+8 and M+9 AMP, ADP, ATP and GMP, GDP, GTP purines, supporting incorporation of serine and glycine derived carbons into purines. Furthermore, UMP, CMP and TMP pyrimidines did not show this increase in RPL10 R98S mutant clones (Figure 4F and Supplementary Figure 10; text lines 222-225). Additionally, we analyzed NAD/NADH,

NAPH/NADPH, and GSH/GSSG ratios, but these were unchanged in RPL10 R98S clones, suggesting that formate is not used to generate reducing equivalents in the mitochondria (see figure below, data not shown in manuscript).

NAD/NADH, NADP/NADPH, and GSH/GSSG ratios in six independent Ba/F3 RPL10 WT clones are compared to five RPL10 R98S clones in two biological repeats.

REVIEWERS' COMMENTS:

Reviewer #2 (Remarks to the Author):

I'm still not convinced that the authors addressed this point. I would suggest to make clear that while the authors present a clear case that in an in vitro system the RPL10 mutation leads to upregulation of PSPH, in a clinical setting it is not clear the role of the mutation as responsible for the changes in serine metabolism.

Reviewer #4 (Remarks to the Author):

I thank the authors for performing these additional experiments to corroborate the activation of serine metabolism in the RPL10-mutant cells. The new data support well the authors' model and have generally improved the quality of the manuscript and I am happy to accept this paper for publication.

One minor point is related to the interpretation of the experiment in figure 5F, and the authors' response to my concern. How do the authors explain that the addition of exogenous serine to the CM of RPL10-mutant cells does not increase the proliferation of normal cells? I would have expected that if the effects of serine on cell proliferation are dose-dependent, they should not be "limited" and that adding more serine to the serine-rich CM of mutant cells would still increase cell proliferation rate. Hence, I suspect that other factors in the CM of mutant cells could explain this increase in proliferation rate.

REVIEWERS' COMMENTS:

Reviewer #2 (Remarks to the Author):

I'm still not convinced that the authors addressed this point. I would suggest to make clear that while the authors present a clear case that in an *in vitro* system the RPL10 mutation leads to upregulation of PSPH, in a clinical setting it is not clear the role of the mutation as responsible for the changes in serine metabolism.

Answer: We show that many T-ALL cases present high PSPH expression, and that reducing PSPH levels reduces the expansion potential of T-ALL cells *in vitro* and *in vivo*. The RPL10 R98S mutation is indeed only one of the mechanisms that can induce PSPH expression in T-ALL, and other unknown mechanisms can also mediate this. As such, the reviewer is correct that the RPL10 R98S mutation alone is not specific to identify patients that may benefit from PSPH inhibition. We have now added 2 sentences in the discussion (lines 391-393) to make this clear.

Reviewer #4 (Remarks to the Author):

I thank the authors for performing these additional experiments to corroborate the activation of serine metabolism in the RPL10-mutant cells. The new data support well the authors' model and have generally improved the quality of the manuscript and I am happy to accept this paper for publication.

One minor point is related to the interpretation of the experiment in figure 5F, and the authors' response to my concern. How do the authors explain that the addition of exogenous serine to the CM of RPL10-mutant cells does not increase the proliferation of normal cells? I would have expected that if the effects of serine on cell proliferation are dose-dependent, they should not be "limited" and that adding more serine to the serine-rich CM of mutant cells would still increase cell proliferation rate. Hence, I suspect that other factors in the CM of mutant cells could explain this increase in proliferation rate.

Answer: Thank you once more for your help on improving our manuscript. With regards to the lack of a dose-dependent effect of serine addition: we believe our results indicate that a dose of 20uM of serine in the medium is already promoting maximal proliferation rates and hence that supplementation of an additional 20uM of serine does not cause an extra effect. Hence the lack of a dose-dependent effect does not necessarily imply that another factor is involved.